# N2M: Bridging Navigation and Manipulation by Learning Pose Preference from Rollout

Kaixin Chai [* 1]    Hyunjun Lee [* 1 2]    Joseph J. Lim [1 3]

## Abstract

Determining where to execute the manipulation policy is a fundamental challenge in mobile manipulation. Most approaches have formulated this as a geometric search problem, prioritizing physical reachability. However, given the high sensitivity of modern learning-based manipulation policies, geometric criteria alone are insufficient. Optimal performance requires base positioning that is aware of the policy's preference. While recent works have attempted to address this, they remain limited in practicality due to reliance on pre-built scene reconstruction and slow inference. In this work, we introduce **N2M** that systematically reformulates the approach to base positioning problem, naturally overcoming limitations of previous methods. Our key insight is that policy preferences are inherent to the local scene structure and can be effectively learned from the policy rollouts. Technically, we propose a novel *viewpoint augmentation* strategy that enables the model to learn robust, viewpoint-invariant pose preferences with remarkable data efficiency. Extensive experiments demonstrate that N2M achieves state-of-the-art performance, outperforming both non-policy-aware baselines and recent policy-aware alternatives. Furthermore, we provide a comprehensive analysis highlighting N2M's broad applicability, generalization capabilities, and data efficiency. Project website: https://clvrai.github.io/N2M/

## 1. Introduction

Mobile manipulators, which integrate mobility and environmental interaction capabilities, hold significant promise for a wide range of real-world applications. By leveraging scene understanding (Rana et al., 2023; Hughes et al., 2022) and navigation modules (Zheng et al., 2025; Chang et al., 2023), these robots can reach the task area based on the task descriptions, and subsequently accomplish the task by executing pre-trained manipulation policies (Intelligence et al., 2025; Kim et al., 2024; Chi et al., 2023).

While this modularity is efficient, it introduces the challenge of determining which pose is suitable to execute manipulation policy. Typically, this is treated as a geometric search. Existing methods (Wu et al., 2025b;a) calculate the geometric feasibility of a grasp, using tools like Inverse Reachability Maps (IRM) (Vahrenkamp et al., 2013), to ensure the target object is physically reachable. However, geometric search is insufficient for modern learning-based manipulation policies due to their extreme sensitivity to initial base poses. A grasp may be kinematically feasible, yet the policy may fail simply because the observation falls outside the training data distribution. While a recent work (Yang et al., 2025) attempt to take training data distribution into consideration, it is less practical due to the reliance on pre-built scene representation and prohibitively slow inference speed.

We introduce N2M (Navigation-to-Manipulation) that systematically reformulates the base positioning problem to be practical and *aware of the policy*, naturally overcoming limitations of previous methods. We demonstrate that our policy-aware approach bridges the critical gap between navigation and manipulation, substantially increasing overall success rates compared to existing works.

We first analyze five fundamental challenges for solving the policy-aware base positioning problem, and propose our corresponding solutions.

**Adaptability to non-static environments**. The environments are typically non-static, requiring predictions to adapt to environmental changes. To support this, N2M predicts the preferable initial pose from the ego-centric RGB point cloud with a single forward pass. This efficient design enables N2M to adjust its predictions in response to environmental changes, as demonstrated in Fig. 12 and 13.

**Multi-modality of preferable initial poses**. Multiple preferable initial poses may exist within the task area. Con-

---
[*]Equal contribution [1]KAIST [2]Seoul National University [3]Vital Robotics. Correspondence to: Kaixin Chai <kaixin-chai@outlook.com>, Hyunjun Lee <hjl1013@snu.ac.kr>.

*Proceedings of the 43rd International Conference on Machine Learning*, Seoul, South Korea. PMLR 306, 2026. Copyright 2026 by the author(s).

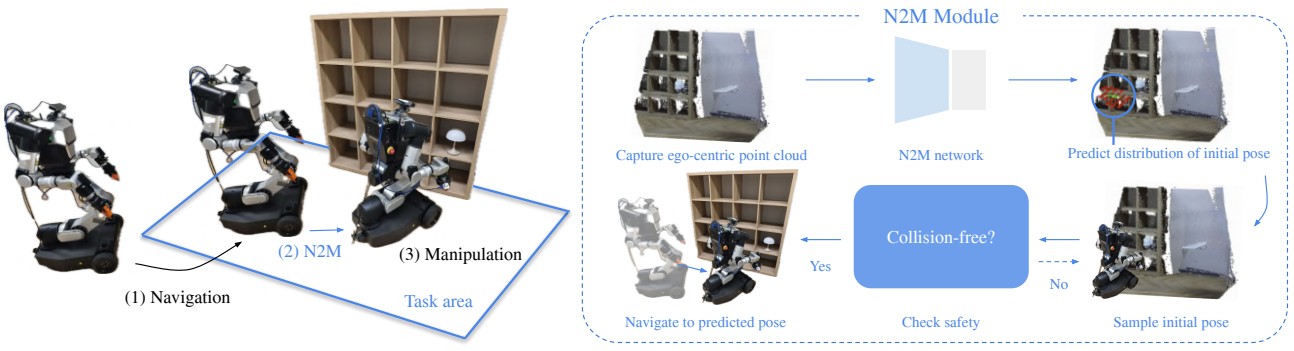

*Figure 1.* System overview. The transition process from the navigation end pose to manipulation initial pose.

sequently, predicting a single pose is insufficient, as it can cause the model to learn an interpolation between viable poses (Bishop, 1994), which may not be preferable to execute manipulation policies. To address this multimodality, N2M predicts the distribution of preferable initial poses, which is represented with a Gaussian Mixture Model (GMM) (Bahl et al., 2023), as shown in Fig. 2.

**Criterion of preferable initial poses**. Manipulation performance depends on multiple factors: policy architecture, training data distribution, robot configuration, task, and environment. Rather than attempting to model these complex relationships, we directly evaluate the pose through policy rollouts. During data collection, we position the robot at various poses and execute the manipulation policy, and successful execution indicates a preferred initial pose. Learning initial pose preferences directly from policy rollouts ensures that N2M's predictions align with the policy's actual performance while enabling broad applicability across diverse policies, tasks, and robot hardware, as shown in Section 5.1.

**Viewpoint Robustness**. Since the robot navigation end poses can be anywhere within the task area, N2M needs to provide reliable predictions at various viewpoints. To achieve this, we augment N2M's training data from multiple viewpoints. Experiments in Sections 5 and 6 demonstrate that N2M reliably predicts preferable initial poses across the whole task area. Interestingly, we note that our proposed data augmentation approach also significantly improves data efficiency and generalizability. We will further analyze the reason behind these benefits in Section 7 and Appendix D.3.

**Data Efficiency**. Collecting rollouts requires substantial time and human effort, as each rollout must be monitored and labeled as success or failure. We incorporate two strategies for data efficiency: First, we design the module to directly predict the initial pose distribution, rather than low-level action (Lee et al., 2019); Second, we augment the dataset through viewpoint rendering to increase its coverage and diversity. In Sections 5.2, 5.3, and 6.1, we demonstrate N2M's remarkable data efficiency and generalizability.

Our contributions can be summarized as follows:

First, we identify and analyze the critical misalignment between navigation and manipulation modules. With experiments, we show that this misalignment stems from the manipulation policy's extreme sensitivity to its initial pose, a factor largely overlooked by prior work.

Second, we propose N2M by systematically redesigning the approach to the robot positioning problem, naturally overcoming the limitations of previous methods. N2M stands out as a highly practical solution, demonstrating broad applicability, real-time performance, viewpoint robustness, remarkable data efficiency, and generalizability.

Third, we propose a novel viewpoint augmentation strategy that enables the model to learn robust and viewpoint-invariant pose preferences with remarkable data efficiency.

Finally, we conduct extensive experiments validating N2M's effectiveness and showing its state-of-the-art performance. We provide detailed quantitative and qualitative results, followed by ablation studies and failure cases, to ensure a comprehensive understanding of our method.

## 2. Related Work

### 2.1. Navigation

Model-based navigation has advanced significantly over the past few decades, enabling mobile manipulation robots to navigate without collisions in unstructured environments (Zheng et al., 2025). The users typically need to explicitly provide the coordinates of the navigation target, which can be obtained by constructing the semantic map (Rosinol et al., 2020) or scene graph (Hughes et al., 2022; Bavle et al., 2023) that associates semantic information with location (Rana et al., 2023). Additionally, RL-based object navigation (Ye & Yang, 2021), or zero-shot navigation based on Large Language Models (LLMs) (Yao et al., 2024) and Vision-Language Models (VLMs) (Zhang et al., 2024b), can also be integrated into the mobile ma-

nipulation system (Chen et al., 2023; Kuang et al., 2024). However, these systems can only determine navigation targets through heuristic rules (Chang et al., 2023; Wang et al., 2023; Liu et al., 2024), such as requiring the robot to face the target object or remain within a specified radius of it. Such heuristics lack the connection to subsequent manipulation policy, often resulting in suboptimal base positioning and failures in manipulation.

## 2.2. Manipulation

Data-driven approaches have demonstrated their advantages in complex and dexterous manipulation tasks. Through experience (Mandlekar et al., 2020; Zhang et al., 2024a) or human demonstrations (Zhao et al., 2023b; Chi et al., 2023), robots can learn manipulation policies. However, due to hardware configuration (Gadre et al., 2022), environmental factors (Abdelrahman et al., 2024), and the distribution of training data (Gao et al., 2024), executing pre-trained policies at different initial poses within the task area yields significantly different success rates. To mitigate this, one line of work aims to enhance the *intrinsic robustness and generalizability* of the policy itself. This is typically achieved by scaling up data collection (Hu et al., 2024) or by leveraging powerful foundation models (Kim et al., 2026). Rather than demanding the policy to master every possible viewpoint, we guide the robot to the specific poses where the policy performs best. Crucially, our method is orthogonal to intrinsic policy generalization; as we demonstrate in our experiments, N2M consistently boosts performance across policies with varying degrees of capability.

## 2.3. Bridging Navigation and Manipulation

The importance of selecting appropriate initial poses for manipulation with mobile robots has long been recognized. Pioneering works addressed it by calculating the Inverse Reachability Map (IRM) (Vahrenkamp et al., 2013; Jauhri et al., 2022), determining initial poses through geometric search. However, while this approaches are sufficient for planner-based policies, it falls short for data-driven policies.

More recent line of works MoManipVLA (Wu et al., 2025b) and MoTo (Wu et al., 2025a) aim to extend off-the-shelf fixed-base manipulation policies with mobile capability. However, similar to the above-mentioned IRM methods, they rely solely on geometric analysis to determine the initial base pose, thereby critically omitting the manipulation policy's sensitivity and preferences. This reliance on geometric methods makes them ineffective for data-driven manipulation policies.

Mobi-$\pi$ (Yang et al., 2025), a concurrent and closest existing work to ours, is currently the only policy-aware method for determining the initial base poses. However, Mobi-$\pi$ suffers from significant practical issues. First, Mobi-$\pi$'s

reliance on the policy's original training data makes the method incompatible with state-of-the-art methods that are often pre-trained on vast datasets (Intelligence et al., 2025; Team et al., 2025) or refined through continuous RL feedback (Amin et al., 2025; Lei et al., 2025). Second, Mobi-$\pi$ incurs a significantly large inference time because necessitating extensive search, iterative sampling, and 3DGS scene reconstruction. This low efficiency inherently restricts its application to static environments. In contrast, N2M is highly practical since it treats the manipulation policy as a black box, learning directly from rollouts without making any assumptions. Moreover, its fast inference enables immediate adaptation to dynamic environmental changes, providing a critical advantage.

## 3. Methodology

### 3.1. N2M module Overview

As illustrated in Fig. 1, our proposed N2M module consists of four steps. First, at the navigation end pose, we capture an RGB point cloud with the RGB-D camera mounted on the robot. Second, our N2M network predicts the distribution of the preferable initial poses from the captured point cloud. Third, a collision-free pose is sampled from the predicted distribution. Finally, the robot navigates to the selected initial pose to execute the pre-trained manipulation policy.

### 3.2. Network Architecture

As the core component of N2M, our network outputs the distribution of initial poses that is preferable for executing manipulation policies. We utilize an RGB point cloud captured from the RGB-D camera mounted on the robot as the input of the network. This design enhances practicality by relying solely on onboard sensors, without any global or historical information during inference.

To effectively capture the multi-modal nature of preferable initial poses within a task area, as shown in Fig. 2, we model the distribution of preferable initial pose $p_\pi$ with GMM:

$$P(p) = \sum_{k=1}^{K} \alpha_k \mathcal{N}(p|\mu_k, \Sigma_k), \tag{1}$$

where $K$ denotes the number of Gaussian kernels, $\alpha_k$ represents the weight of the $k$-th kernel, and $\mathcal{N}(p|\mu_k, \Sigma_k)$ signifies the Gaussian distribution with mean $\mu_k$ and covariance matrix $\Sigma_k$.

Our N2M network, $f_\theta$, predicts the parameters $\{\alpha_k, \mu_k, \Sigma_k\}_{k=1}^{K}$ using RGB point cloud observation $o$, captured by an onboard RGB-D camera. As illustrated in Fig. 3, the point cloud is encoded by Point-BERT (Yu et al., 2022) into a fixed-length latent vector, which then passes through a multi-layer perceptron (MLP) to generate

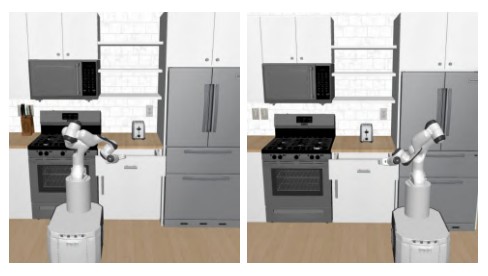

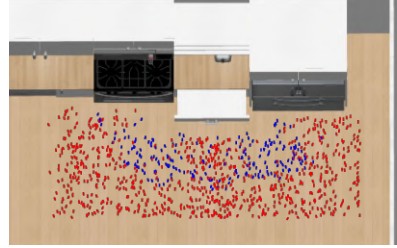

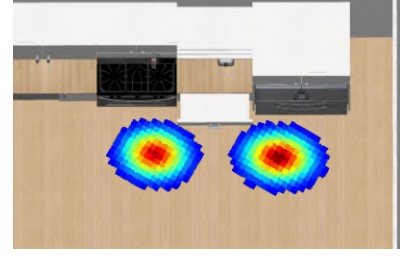

(a) Robocasa Close Drawer Task

(b) Rollout Results

(c) Predicted Preferable Initial Pose Distribution

*Figure 2.* (a) Preferable initial poses of the *Close Drawer* task are inherently multi-modal, as the manipulation policy is learned to close drawers from both sides. (b) visualizes preferable initial poses from successful rollouts (blue), which shows multi-modality. (c) With GMM, we effectively model this multimodal distribution of preferable initial poses.

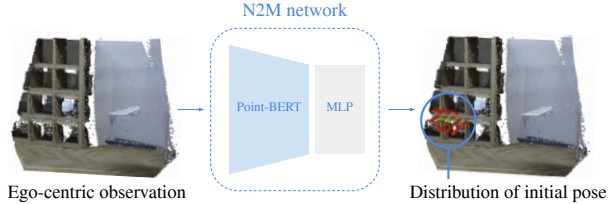

*Figure 3.* N2M network takes ego-centric RGB point clouds to predict the distribution of preferable initial poses.

the parameters for each Gaussian kernel of the GMM.

The network is trained to maximizes the probability of preferable initial poses,

$$L(\theta) = \sum_{(o,p) \in D} -\log P_{f_\theta(o)}(p), \quad (2)$$

where $D$ denotes the dataset consisting of observation–pose pairs, and $(o, p)$ represents the element in the dataset. We fine-tune the pre-trained Point-BERT along with MLP layers as we empirically find that this leads to better performance.

### 3.3. Data Preparation

Preparing training data for the N2M network involves two steps: collecting the raw dataset $R$ and augmenting it to create the training dataset $D$.

#### 3.3.1. RAW DATA COLLECTION

The raw dataset $R$ consists of entries $(S, p_\pi)$, where $S$ represents a local scene reconstruction and $p_\pi$ denotes a preferable initial pose for manipulation policy execution, as illustrated in Fig. 4(a).

For each entry, the collection process proceeds as follows:

1. Multiple RGB point cloud frames are captured and stitched together to reconstruct the local task area $S$, with their relative poses determined through odometry (Mohamed et al., 2019) or point cloud registration (Huang et al., 2021).

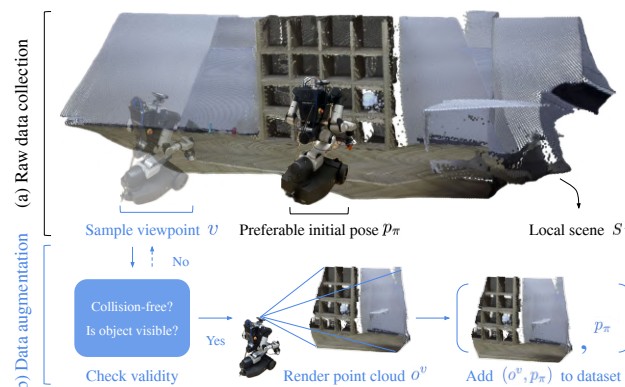

*Figure 4.* Data preparation process: (a) Raw data collection, showing scene $S$ and preferable pose $p_\pi$; (b) Data augmentation, rendering scene from diverse viewpoints.

2. A pose within the task area is selected for policy rollout. If the manipulation policy $\pi$ successfully completes the task, this pose is recorded as $p_\pi$ for the current scene $S$. The scene is then randomly reset for the next rollout.

Note that $p_\pi$ and $S$ must share the same coordinate frame. However, the specific choice of reference frame does not matter, as we will transform the coordinate during both training and inference to the body coordinate of the robot.

#### 3.3.2. DATA AUGMENTATION

N2M is designed to predict the distribution of $p_\pi$ based on ego-centric observations. Since the navigation end pose can be anywhere within the task area, we apply data augmentation to enhance N2M's robustness to viewpoint variations.

As shown in Fig 4(b), for each collected scene-pose pair $(S, p_\pi)$, we first uniformly sample $M$ different viewpoints within the task area. We then filter out viewpoints that either collide with the scene or from which the object is not visible. For each verified viewpoint, $v$, we render point cloud $o^v$ by projecting points from $S$ to the viewpoint using the intrinsics of the RGB-D camera mounted on the robot. Preferable initial pose $p_\pi$ for a given scene $S$ remains invariant across viewpoints. Therefore, all rendered observations $o^v$ from the

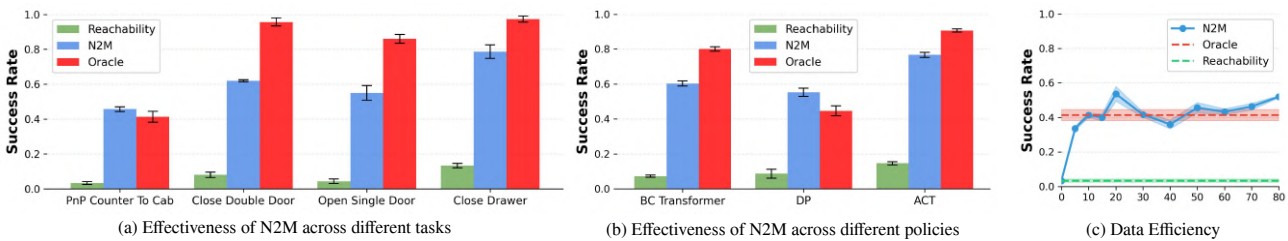

(a) Effectiveness of N2M across different tasks     (b) Effectiveness of N2M across different policies     (c) Data Efficiency

*Figure 5.* Simulation results demonstrating N2M's broad applicability and data efficiency. The plots show the success rates across (a) diverse tasks, (b) different policy architectures, and (c) varying numbers of training rollouts.

same scene share the same label $p_\pi$, and each pair $(o^v, p_\pi)$ is added to the training dataset $D$.

## 4. Experiment Settings

### 4.1. Objectives

Our experiments in Sections 5 and 6 are designed to answer the following core questions: **Q1 (Applicability):** Can N2M capture pose preferences across diverse policies and tasks? **Q2 (Data Efficiency):** Can N2M learn robust preferences from sparse rollouts? **Q3 (Generalizability):** Do learned preferences transfer zero-shot to unseen scenes? **Q4 (Benchmark):** Does N2M outperform baselines in both success rate and inference speed? **Q5 (Performance in Real World):** Do our findings in the simulation still hold in the real world? **Q6 (Sequential Amplification):** How does N2M mitigate the error compounding inherent in sequential tasks, thereby amplifying overall task success?

### 4.2. Environments

**Simulation** We conduct experiments in RoboCasa (Nasiriany et al., 2024), a simulation framework that provides diverse manipulation tasks along with pre-collected demonstrations for policy training. Details are provided in Appendix B.

**Real world** We manually design 2 real world tasks to demonstrate N2M's comprehensive performance in generalization and sequential tasks. Details can be found in Appendix C.

### 4.3. Baselines

**Reachability** A non-policy-aware method that only guarantees physical reachability. It is commonly used in many work like MoManipVLA (Wu et al., 2025b) and MoTo (Wu et al., 2025a), and IRM (Vahrenkamp et al., 2013; Jauhri et al., 2022).

**Mobi-$\pi$ (Yang et al., 2025)** The only existing policy-aware method alternative to ours. Mobi-$\pi$ uses sampling-based optimization to identify the optimal robot pose in a pre-build 3DGS-based scene reconstruction.

**Oracle** An idealized baseline where the manipulation policy is evaluated at a fixed, pre-defined pose identical to the one used during data collection in RoboCasa. This represents the upper bound of performance.

**Human** A baseline that human intuition is use to determine a pose that is not only geometrically feasible but also intuitively suitable for manipulation. We recruited participants who were informed of the task objectives but were blinded to the data collection process. They selected poses based on their intuitive judgment.

## 5. Simulation Experiment

### 5.1. Evaluation of Applicability (Q1)

Since N2M doesn't rely on any assumptions about tasks, policies, and robot hardware, it has a broad applicability. To validate this, we select four predefined tasks and three policy designs in RoboCasa. We train multiple N2M modules using 5 to 70 successful rollouts and select the best-performing model for each setting. We report the averaged success rate from 300 trials in Fig. 5.

N2M consistently outperforms the reachability baseline, demonstrating the limitation of non-policy-aware methods. This emphasizes the necessity of policy-aware methods that can effectively bridge navigation and manipulation. Second, the success rate with our N2M module is comparable to the oracle baseline, indicating that N2M can reliably estimate preferable initial poses across various settings.

Notably, for the *PnPCounterToCab* task in Fig 5(a) and DP in Fig 5(b), N2M outperforms the oracle baseline. This is especially remarkable, as it demonstrates that the policy's preference does not necessarily align with the distribution from training data. N2M module, trained directly from policy rollouts, effectively captures these preferences, achieving superior performance compared to the oracle baseline. This finding wouldn't have been possible with similarity-based in-distribution estimation methods, highlighting the importance of learning from rollouts that reflect the actual behavior and preference of the manipulation policy.

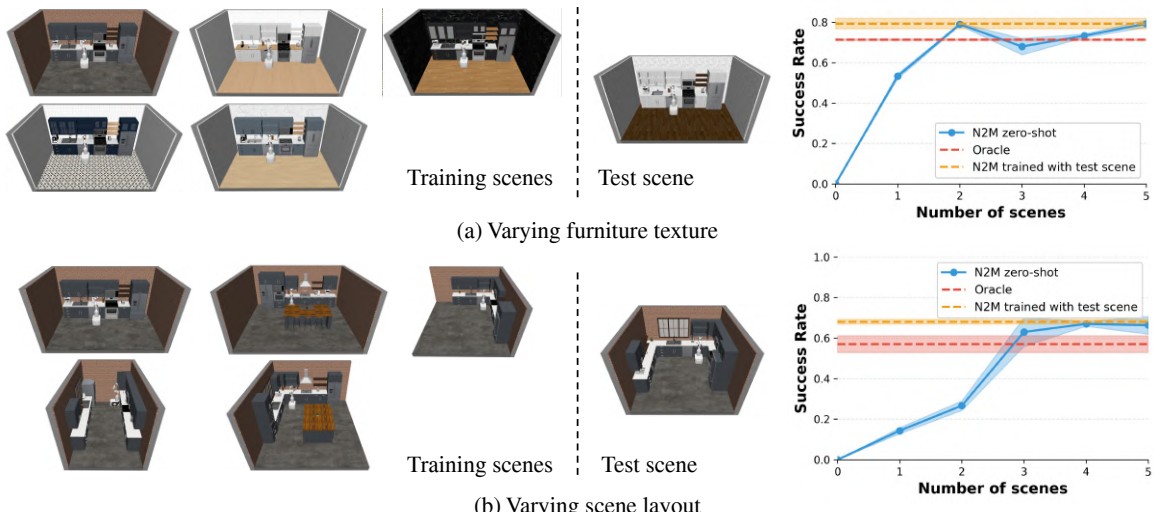

(a) Varying furniture texture

(b) Varying scene layout

*Figure 6.* Analysis of N2M's generalizability. Success rate improves significantly as the number of training scenes increases.

## 5.2. Data Efficiency Analysis (Q2)

We further analyze the data efficiency of N2M using the *PnPCounterToCab* task. We evaluate the averaged success rate of N2M modules trained with varying numbers of rollouts in N2M's training. In each rollout, the apple's position, color, and shape vary while the kitchen furniture remains consistent. The module is then tested in the same scene. As shown in Fig 5(c), the averaged success rate of the manipulation policy matches the oracle baseline with only 10 rollouts and even surpasses it with 20. Although some fluctuations indicate sensitivity to sample variations, the overall trend shows that N2M effectively captures the policy's preference with a small number of rollouts.

## 5.3. Generalizability Analysis (Q3)

We evaluate the generalizability of the N2M module in the *PnPCounterToCab* task based on the number of distinct scenes used to collect successful rollouts for training. We vary the number of training scenes from 0 to 5, collecting 10 successful rollouts in each scene, resulting in a total of 0, 10, 20, 30, 40, and 50 rollouts, respectively. The trained module is then tested in an unseen scene. We design two groups of varying scenes. For the first group, as shown in Fig 6, we vary the furniture texture while keeping the kitchen layout fixed, whereas for the second group, we vary the furniture layout while keeping the furniture texture fixed.

Graphs in Fig. 6 demonstrates that the N2M module can effectively estimate the initial pose preference of the manipulation policy even in unseen environments. As we increase the number of scenes for rollout collection, the module's performance improves accordingly, matching and even surpassing the oracle baseline. This result indicates that N2M can capture the general pattern of both the tasks and corresponding manipulation policies with a small number of

scenes and apply the learned pattern in unseen scenarios.

## 5.4. Benchmark (Q4)

To further illustrate the distinctions between N2M and existing methods, we conducted benchmark experiments. We selected the CloseDrawer task, which is shared between Mobi-$\pi$ (Yang et al., 2025) and our work, and tested with two different policies: BC Transformer (trained following RoboCasa's official guide and open-source data) and a diffusion policy (using Mobi-$\pi$'s open-source configuration and checkpoint). We performed 6 independent experiments, each consisting of 50 inference trails, to calculate the success rates $S_{DP}$ and $S_{BC}$ for all methods under both policies. Detailed benchmark settings can be found in Appendix B.6

| Method | $S_{\text{DP}}$ | $S_{\text{BC}}$ | $T_{\text{human}}$ | $T_{\text{inference}}$ |
|---|---|---|---|---|
| Oracle | 0.93±0.04 | 0.97±0.02 | - | - |
| Reachability | 0.16±0.04 | 0.15±0.03 | 0 | 0.72±0.36 s |
| Mobi-$\pi$ | 0.24±0.05 | 0.09±0.03 | $n\times5$ min | 273.52±13.06 s |
| N2M | **0.55±0.07** | **0.56±0.04** | 20.7 min | **0.07±0.03** s |

*Table 1.* Performance comparison with baselines. Quantitative comparison of success rates and time costs. We report the success rates $S_{\text{DP}}$ and $S_{\text{BC}}$ under two different policies, alongside human labor $T_{\text{human}}$ and inference time $T_{\text{inference}}$.

**Success Rate Analysis:** Reachability-based methods show poor performance as they are not aware of policy's pose preferences. Surprisingly, Mobi-$\pi$ doesn't perform well under either policy. We attribute this to two factors: first, observation similarity is just a heuristic of policy performance and cannot indicate the true policy performance; second, Mobi-$\pi$ uses a pretrained RGB encoder to extract RGB features, which might not be capable of distinguishing minor observation differences, as also discussed in their paper. In contrast, N2M learns pose preferences directly from policy rollouts, enabling more accurate prediction of preferred

base positions and achieving state-of-the-art performance compared to existing methods.

**Time Cost Analysis:** Mobi-$\pi$ requires prior scene reconstruction (5min scanning + 7min 3DGS training), and Bayesian optimization during inference (273.52s each). Users have to reconstruct the scene again for every environmental change. We use $n$ to represent the number of changes. In contrast, N2M requires policy rollout collection (20.7min) and training (3.4 hours), but achieves rapid inference (0.07s each), enabling real-time prediction in dynamic environments.

In summary, N2M accurately predicts the preference of manipulation policy with real-time performance. In contrast, alternative approaches either suffer from low performance (reachability method) or slow inference speeds (Mobi-$\pi$). While N2M requires data collection, our proposed viewpoint augmentation technique substantially reduces the number of policy rollouts needed, making the approach more practical for real-world deployment.

## 6. Real-World Experiment

### 6.1. Apply N2M to Real-world Task (Q5)

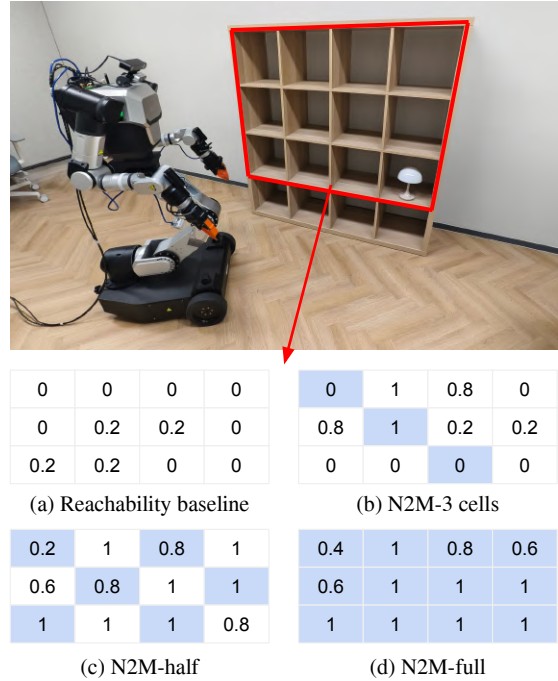

| 0 | 0 | 0 | 0 |
|---|---|---|---|
| 0 | 0.2 | 0.2 | 0 |
| 0.2 | 0.2 | 0 | 0 |

(a) Reachability baseline

| 0 | 1 | 0.8 | 0 |
|---|---|---|---|
| 0.8 | 1 | 0.2 | 0.2 |
| 0 | 0 | 0 | 0 |

(b) N2M-3 cells

| 0.2 | 1 | 0.8 | 1 |
|---|---|---|---|
| 0.6 | 0.8 | 1 | 1 |
| 1 | 1 | 1 | 0.8 |

(c) N2M-half

| 0.4 | 1 | 0.8 | 0.6 |
|---|---|---|---|
| 0.6 | 1 | 1 | 1 |
| 1 | 1 | 1 | 1 |

(d) N2M-full

*Figure 7.* The *Lamp Retrieval* task with averaged success rates in each cell. The 3×4 table represents the top three rows and all four columns of the shelf. We collect one successful rollout per cell colored in blue to train N2M.

We choose the *Lamp Retrieval* task shown in Fig 7, and evaluate three variants of N2M module along with the reach-ability baseline. In Fig. 7, we mark the cells where the rollouts are collected in blue, along with the success rate out of five trials. We collect one rollout for each marked cell, resulting in 3, 6, and 12 rollouts for the N2M-3 cells, N2M-half, and N2M-full variants, respectively.

As shown in Fig 7(a), the performance of the reachability baseline is notably low, indicating the limitation of non-policy-aware methods. Fig. 7(b-d) shows that N2M effectively predicts preferable initial poses with only a small amount of rollouts, showcasing the data efficiency of our method. Notably, Fig. 7(b) and (c) further illustrate the generalizability of our approach: although rollouts are collected from a subset of cells, N2M can give reasonable predictions even when the lamp is placed in the cells where the rollouts are not collected.

To test the viewpoint robustness and reliability of N2M, we demonstrate ten consecutive successful task executions, as shown in Fig. 12, with the N2M module trained using 12 rollouts. Before each execution, the lamp was randomly placed in one of the cells among the top three rows of the shelf, and the robot was randomly initialized within a $2 \times 3$ m area in front of the shelf, regarded as the navigation end pose in the task area. The robot's orientation is also randomized, but we ensure that the lamp remains visible to the RGB-D camera.

### 6.2. Performance in Sequential Task (Q6)

Optimizing the transition between navigation and manipulation is particularly critical in sequential task scenarios. We designed the multi-stage task illustrated in Fig. 8, which requires the robot to: (1) navigate to a table, (2) grasp an empty chip box, (3) navigate to a trash bin, and (4) drop the chip box into the trash bin.

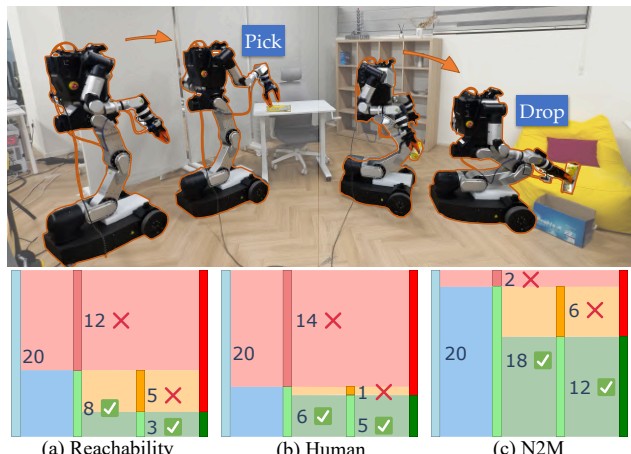

(a) Reachability   (b) Human   (c) N2M

*Figure 8.* Task success rates across different methods in a sequential task. We use *Sankey diagrams* to visualize the flow of success through the **Pick** and **Drop** stages.

We evaluated this multi-stage task over 20 trials, comparing

three methods: Reachability, Human-Intuition, and N2M. The poses of the table, trash bin, and chip box were randomized in each trial. The results, presented in Fig. 8, yield two critical insights:

First, reachability alone is insufficient for effective manipulation. Even human operators struggle to determine the optimal base pose without prior knowledge of the policy. Notably, humans begin to discern the policy's spatial preferences after several trials, gradually improving their success rates. This observation validates the design intuition behind N2M, which similarly learns these preferences through policy rollouts.

Second, the sequential nature of the task demonstrates how performance compounds across subtasks, making the optimization of base positioning critical at every stage. By capturing specific policy preferences, N2M substantially improves both subtask success and overall sequence completion rates. This underscores the necessity of policy-aware positioning in long-horizon mobile manipulation.

## 7. Further Analysis

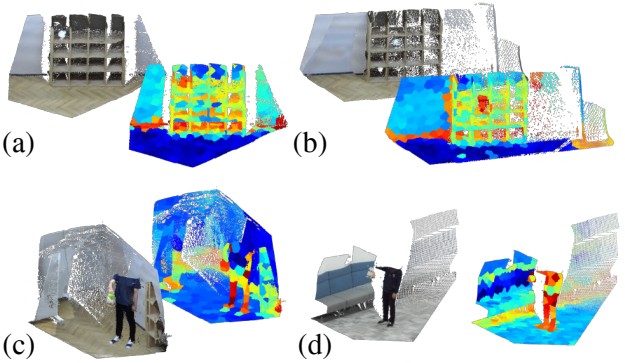

*Figure 9.* Visualization of learned representations from the N2M encoder. (a, b) For the *Lamp Retrieval* task, the encoder focuses on the lamp, while (c, d) for the *Toybox Handover* task, the encoder focuses on the person and the toybox. Notably, (d) highlights the encoder's ability to identify salient regions in unseen environments.

**Qualitative Analysis** We qualitatively analyze that N2M is capable of generating predictions in real-time, show 10 consecutive success from varying viewpoints showing viewpoint robustness, and show that N2M can generalize to totally unseen environments even with few data. Refer to Appendix D.1 for more details.

**Ablation Study** We ablate the core design choices of N2M. Our results show that viewpoint augmentation and positional augmentation contributes most to the success of N2M while the encoder architecture and the use of pretrained weights matters less. We further ablate the effect of the number of viewpoints to augment ($M = 300$) and show that the success rate increases as we augment with more viewpoints

and saturates near the number 300. Refer to Appendix D.2 for more details.

**Visualization of Learned Representations** We further visualize the learned representation of the encoder to analyze the success of N2M. With the encoder's output features, we highlighted the region that the model focuses on. As shown in Fig 9, the model consistently focuses on the region of interest, such as lamp, toybox, and the person holding the object. Remarkably, Fig 9(d) demonstrates robust generalization to unseen environments with less than 20 rollouts. Even though the background and the person differ from the training data, the model still identifies the toybox and the person holding it. This indicates that N2M learns to reliably capture salient regions in a highly data-efficient manner. Refer to Appendix D.3 for more details and results.

**Failure Analysis** To provide a comprehensive understanding of N2M, we analyze its typical failure patterns. We observed that N2M is sensitive to small objects and far distances. Furthermore, performance relies heavily on sensor quality. To mitigate this issue, we employed a ZED2i camera to ensure high-fidelity point clouds. Finally, the module is intrinsically linked to the manipulation policy's performance. Even when N2M predictions are accurate, task failure cannot be prevented if the policy distribution of preferred initial poses is too narrow. See Appendix D.4 for further details.

## 8. Conclusion and Discussion

In this work, we reformulate the base positioning problem from a geometric search into a preferable pose prediction task supervised by rollout outcomes. This paradigm shift enables N2M to significantly outperform existing methods in both inference speed and success rates. Technically, we propose a novel *viewpoint augmentation* strategy that allows the model to learn viewpoint-invariant representations, ensuring consistent predictions regardless of the robot's perspective, a technique that can be generalized to other 3D scene prediction domains.

In a broader sense, the base pose distribution, predicted by N2M, serves as an indicator of the policy proficiency. Unveiling the policy performance boundary offers a promising avenue for guiding active learning, enabling targeted data collection to continuously extend policy capabilities.

Finally, it is worth noting that N2M improves policy performance by selecting preferable base poses, which is orthogonal to enhancing intrinsic policy capabilities. On the one hand, N2M consistently boosts success rates regardless of the policy's inherent capability. On the other hand, unlike resource-intensive methods relying on data scaling or prowerful but heavy backbones, N2M offers a plug-and-play enhancement with negligible computational overhead.

## Acknowledgements

This work was supported by Institute of Information & communications Technology Planning & Evaluation (IITP) grant (No.RS2019-II190075, Artificial Intelligence Graduate School Program, KAIST; No.2022-0-00077, AI Technology Development for Commonsense Extraction, Reasoning, and Inference from Heterogeneous Data; No.RS-2022-II220984, Development of Artificial Intelligence Technology for Personalized Plug-and-Play Explanation and Verification of Explanation), National Research Foundation of Korea (NRF) grant funded by the Korea government (MSIT) (NRF-2021H1D3A2A03103683, Brain Pool Research Program; RS-2024-00414822), and the Technology Innovation Program(or Industrial Strategic Technology Development Program-Robot Industry Technology Development)(RS2024-00427719, Dexterous and Agile Humanoid Robots for Industrial Applications) funded by the Ministry of Trade Industry & Energy(MOTIE, Korea)

The authors are deeply grateful to Jeongjun Kim, Sunwoo Kim, Junseung Lee, Doohyun Lee, and Minho Heo for their insightful discussions and continuous support throughout this project. We also sincerely thank RAINBOW ROBOTICS for their generous hardware support, which made our real world experiments possible.

## Impact Statement

This paper presents work whose goal is to advance the field of Machine Learning. There are many potential societal consequences of our work, none which we feel must be specifically highlighted here.

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

# A. Training Details

## A.1. Data Augmentation

In addition to the viewpoint augmentation described in Sec 3, we apply two further augmentations during training to improve the robustness of our module. First, we perform random rotations around the Z-axis and translations within a $1m$ radius circle on the XY-plane. Second, we uniformly downsample the point cloud to 8,192 points, following the original Point-BERT setting.

## A.2. Regularization Term of Loss Function

To better fit the distribution of preferable initial poses with GMM, we introduce three additional regularization terms. First, we maximize the entropy ($\mathcal{H}_w = -\sum_i w_i \log w_i$) of kernel weights to discourage the model from collapsing into a single mode. Second, we enforce inter-mode distance ($\mathcal{D} = \sum_{i<j}(\mu_i - \mu_j)^T \Sigma_{\text{avg}}^{-1}(\mu_i - \mu_j)$) where $\Sigma_{\text{avg}}$ is the average of covariance matrix) to prevent different components from converging to the same distribution. Finally, we regularize the weighted sum of entropy of each modes ($\mathcal{H}_{\text{mode}} = \sum_i w_i \mathcal{H}_i$ where $\mathcal{H}_i$ is the entropy of $i^{\text{th}}$ mode) to avoid overfitting by ensuring each mode does not become overly narrow. In summary, the loss function is as follows:

$$L(\theta) = \sum_{(o_i,p_i)\in D} -\log P_{f_\theta(o_i)}(p_i) \; -\alpha_w \mathcal{H}_w - \alpha_{\text{dist}}\mathcal{D} - \alpha_{\text{mode}}\mathcal{H}_{\text{mode}}. \tag{3}$$

# B. Detailed Settings for Simulation Experiment

## B.1. Task and Policy

We choose four tasks, as shown in Fig. 10(a-d): (a) **PnPCounterToCab**. Pick an apple from the counter and place it in the cabinet (b) **Close Double Door**. Close the cabinet doors on both the left and right sides. (c) **Open Single Door**. Open a microwave oven. (d) **Close Drawer**. Close a drawer. We got rid of the distracters during the environments. For the *PnPCounteToCab* task, we randomized the shape, color, and position of the apple. Except for the generalizability experiment in Section 5.3, all experiments were conducted in a single environment without changing the furniture texture and layout both during rollout collection and N2M inference.

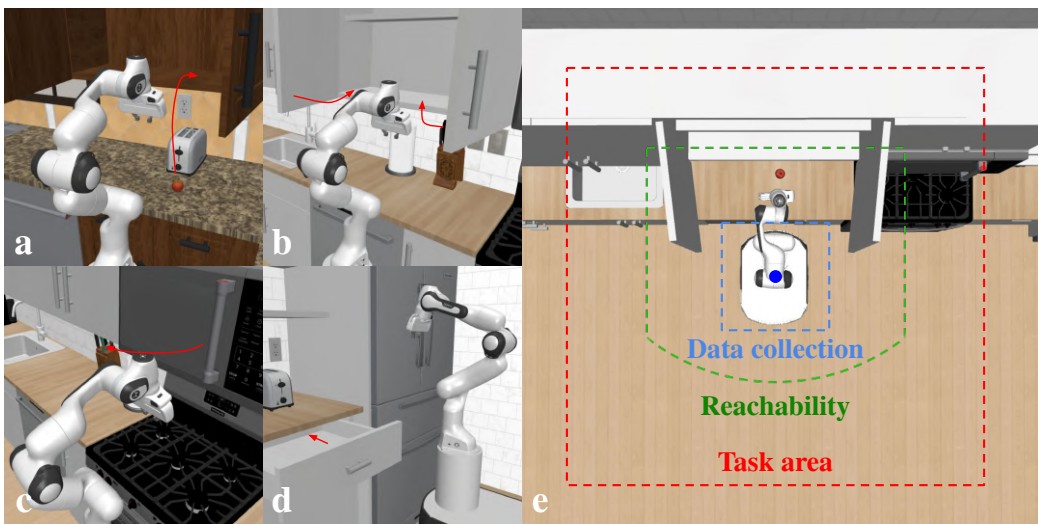

*Figure 10.* Task and randomization criterion in Simulation Experiment.

In Section 5.1, we train BC Transformer (Mandlekar et al., 2021) across all tasks to compare the performance across tasks. For comparison between policies, we train three different policies: BC Transformer, Diffusion Policy (DP) (Chi et al., 2023), and Action Chunking with Transformers (ACT) (Zhao et al., 2023a) in the *Open Single Door* task. We train each manipulation policy with 3000 demonstrations provided in RoboCasa.

To predict the distribution of the preferable initial pose of the policy, we use two kernels ($K = 2$) for the Close Drawer task as the distribution is expected to have two modes, and a single kernel ($K = 1$) for other tasks.

## B.2. Randomization Criterion

We introduce three randomization criteria for initializing the robot pose. Note that the demonstrations provided by RoboCasa are collected from a fixed pose, and we define the randomization region based on a square centered at this reference pose.

**N2M Data collection randomization** $0.4 \times 0.4$ m square centered at the reference pose with $15°$ angular variance. Used for collecting successful rollouts to train N2M network.

**Reachability randomization** Intersection of $1 \times 1$ m square centered at the reference pose and a circle with a 1m radius centered at the target object with $30°$ angular variance. This setup captures feasible base poses for naive navigation-to-manipulation transitions based on the robot arm length.

**Task area randomization** $2 \times 2$ m square centered at the reference pose with $30°$ angular variance. An additional constraint is imposed, requiring the target object to be visible from the given pose. The region indicates navigation end poses where we capture the RGB point cloud for N2M inference.

## B.3. Collision Detection

To sample collision-free poses from the distribution predicted by N2M, we project the scene's point cloud onto a 2D plane to serve as an occupancy map. We then perform collision checking by modeling the robot's footprint as a rectangular boundary. For point cloud acquisition, we utilize the same camera configuration established during the data collection phase.

## B.4. Robot Setup

We use a Franka Panda arm mounted on an Omron mobile base, with an additional RGB-D camera attached to the robot's wrist to capture an ego-centric point cloud. We use the ground truth depth and robot location, allowing perfect reconstruction of the point cloud.

We fix the initial joint configuration across all tasks, allowing us to decouple joint positions from the robot's base pose and predict policy preference solely in SE(2) space.

## B.5. Implementation of Robot Transition

During evaluation, after N2M prediction, we utilize MuJoCo's API to place the robot at that predicted pose for efficient simulation. We also implement a simple motion-planning algorithm for the differential-drive base to facilitate natural visualization.

## B.6. Detailed Settings in Benchmark Experiment

In this section, we provide a comprehensive description of the computational platform used for the benchmark experiments and further elaborate on the temporal statistics regarding data collection, training, and inference.

**Hardware Configuration** All experiments in our benchmark were conducted on a workstation equipped with an NVIDIA GeForce RTX 3090 (24GB) GPU and an AMD EPYC 7763 64-Core Processor. All reported durations represent wall-clock time. While absolute training and inference latencies may vary across different hardware platforms, we expect the relative computational overhead between methods to remain consistent.

**Deployment of Rechability Baseline** The implementation of the reachability baseline in our benchmark slightly differs from the version described in other sections of this paper, although this discrepancy does not yield significant differences in the final results. Specifically, in other parts of the paper, we simplify the deployment by assuming that the reachability condition is satisfied if the target object is within the workspace radius of the robotic arm. In contrast, to ensure a more rigorous and nominal comparison within the benchmark, we employ the Pinocchio library to perform Inverse Kinematic (IK) calculations and comprehensive collision checks based on the precise spatial coordinates of the objects. This high-fidelity baseline provides a more formal validation of feasibility compared to the distance-based heuristic. We again emphasize that the implementation of high-fidelity Inverse Kinematic (IK) calculations did not yield significant performance differences compared to the simplified randomization criteria employed in our other experiments. This consistency validates our use of

the randomization criterion as a reliable and computationally efficient proxy for the reachability baseline.

**Time Cost of Mobi-**$\pi$ Following the official implementation guidelines for Mobi-$\pi$, we captured 1,000 images within the simulation to perform 3D Gaussian Splatting (3DGS) reconstruction. This process is nearly instantaneous in simulation as it does not account for physical robot movement. To reflect realistic constraints, we assume a duration of 0.3 seconds per valid image, totaling 5 minutes per scene for data acquisition. The 3DGS model training requires an additional 7 minutes. Post-training, the **single-inference** time is $273.52 \pm 13.06$ seconds. This prolonged inference is due to their strategy of dense sampling within the region for iterative Bayesian optimization, which necessitates continuous calls to the 3DGS renderer for camera observations at each sampled pose. These statistics were derived using the authors' official codebase.

**Time Cost of N2M** We standardized the data collection process to 20 successful rollouts. It should be noted that the total duration accounts for all attempts, including failed rollouts, until the target count of 20 successes is reached. For the *open drawer* task, data collection within a $0.2m \times 0.2m \times 10°$ region took 12.9 minutes, including 6 failed attempts. Since the simulator does not explicitly model scene capture or environment reset times, we assume the following overheads: capturing 4 point clouds per scene at 2 seconds each, and a 10-second reset period per rollout. This introduces an additional 7.8 minutes of overhead calculated as $(20 + 6) \times (2 \times 4 + 10)$s, resulting in a total collection time of 20.7 minutes. We also recorded the training time for N2M, which totals 3.4 hours including the fully automated viewpoint augmentation process. Once trained, the model achieves a rapid single-inference time of $0.07 \pm 0.03$ seconds.

## C. Detailed Settings for Real-world Experiment

### C.1. Task and Policy

For real-world scenarios, we designed five tasks, as shown in Fig. 11(a-e): (a) **Lamp Retrieval**. The lamp is randomly placed in one cell among the top 3 rows of a shelf, 12 cells in total, with variations of up to 3cm within each cell. (b) **Open Microwave**. The robot should open a microwave that is randomly placed on a white table. (c) **Use Laptop**. The robot should use a laptop that is randomly placed on a black round table, and the table is randomly placed in a room. (d) **Push chair**. The robot should push a chair that is randomly placed in a room. (e) **Toybox Handover**. The robot should take a toybox from a person randomly standing in the room and holding a toybox at varying heights.

For the manipulation policy in task (a), we collect 50 demonstrations from each cell with base randomized within a $0.2 \times 0.2$ m square region with angular variance $\pm 60°$. This results in a total of 600 demonstrations, which are then used to fine-tune $\pi_0$ (Black et al., 2024).

For the N2M module, we use a single kernel ($K = 1$) across all the tasks in the real world.

### C.2. Randomization Criterion

Specifically, for task (a), we test the N2M module with an actual policy and evaluate the success rate. For tasks (b)-(e), we test the N2M module without policy by manually labeling and gathering positive rollouts based on human-defined rules. With this setup, we demonstrate N2M's high data efficiency, generalizability, and real-time adaption to the dynamic environments.

In real-world experiments, we adopt a different randomization strategy for N2M data collection, reachability randomization, and task area randomization.

**N2M Data collection randomization** We manually pick candidates of the initial pose in the task area to collect successful rollouts, which is more efficient.

**Reachability randomization** Intersection of $0.5 \times 0.5$ m rectangular region centered 0.5m away from the object and a circle with radius 1m centered at the object with angular variance $60°$. This represents the region within

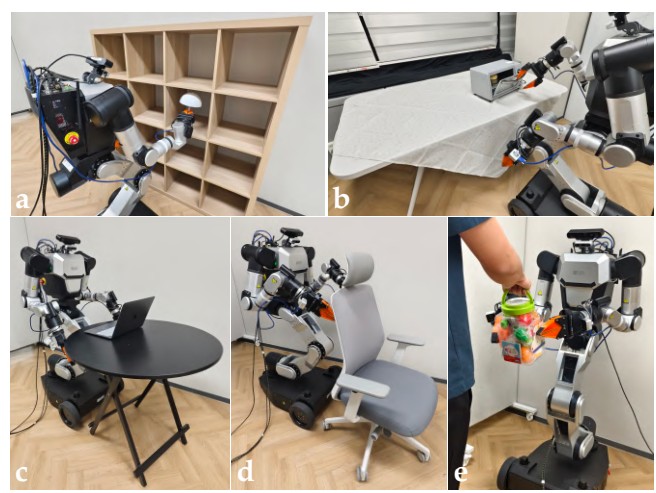

*Figure 11.* (a) Lamp Retrieval (b) Open Microwave (c) Use Laptop (d) Push Chair (e) Toybox Handover. Tasks (b-e) are for further analysis in Appendix D.1.3

the robot's reach, given the arm length is around one
meter.

**Task area randomization** We utilize the whole room to randomize the base pose with additional constraint that the object should be visible. Following simulation experiment, this also indicates navigation end pose where we capture RGB point cloud for N2M inference

### C.3. Collision Detection

Following our simulation setting, we project the scene's point cloud onto a 2D plane to generate an occupancy map and model the robot's footprint as a rectangular boundary for collision checking. In real-world experiments, however, we derive this map from local point clouds captured by the ego-centric camera. While this approach relies on partial scene observation rather than global context, it proves sufficient for base pose prediction, as both the target object and the preferable base pose are typically captured within the camera's field of view.

### C.4. Robot Setup

For real real-world experiment, we employ the Rainbow Robotics RB-Y1 robot[1] platform. We use three cameras in total: a RealSense D405 camera on the right wrist of the robot, a RealSense D435, and a ZED 2i camera on the head of the robot. We use RealSense cameras for manipulation policy and the ZED 2i camera to capture the ego-centric RGB point cloud of the scene. We utilize two 2D LiDAR sensors attached to the robot base to get the odometry.

As the RB-Y1 robot offers height adjustment, we incorporate torso height into the robot's initial pose. Following the simulation setup, we fix the initial joint configuration of the robot arm, allowing us to decouple joint positions from the initial pose. As a result, the robot's initial pose is represented as a 4-dimensional vector $(x, y, \theta, h)$.

### C.5. Implementation of Robot Transition

We implement a simple motion-planning algorithm for the differential-drive base to transit the robot from the end pose of navigation to the predicted initial pose for executing the manipulation policy. Although it does not consider collisions, it is sufficient for our experiments, as motion planning is not the primary focus of our work.

### C.6. Details of Comprehensive Case 3

**Randomization Criterion.** The trash bin and table positions are randomized within a 3m × 5m area for each trial. The chip box is randomly placed on the table surface (40cm × 80cm).

**Policy Training.** We collect 200 demonstrations for each task (picking and disposal) which are used to fine-tune the pre-trained $\pi_0$ policy.

**N2M Data Collection and Training.** For each trained manipulation policy, we conduct 20 rollouts with randomized scene configurations. We apply viewpoint augmentation with M=300 and train the N2M network for 150 epochs.

**Human-intuition Baseline.** We recruit two participants to determine robot base pose for policy execution. Participants are informed only about the task objectives but not about policy training methodologies or data collection strategies. Robot positioning is based solely on their intuitive judgment of optimal placement.

**Reachability Baseline.** We utilize AprilTags for object pose estimation and filter out infeasible robot base pose by collision checking and IK-based reachability analysis to ensure the target object is within the manipulator's workspace.

## D. Further Analysis Details

### D.1. Qualitative Analysis

This section provides detailed visualizations and experiments supporting the qualitative claims made in Section 7 regarding robustness, real-time adaptability, and generalization.

---

[1]https://www.rainbow-robotics.com/rby1

### D.1.1. VIEWPOINT ROBUSTNESS

As shown in Fig. 12, we show ten consecutive successes of the *lamp retrieval* task. Before each execution, the lamp was randomly placed in one of the cells among the top three rows of the shelf, and the robot was randomly initialized within a $2 \times 3$ m area in front of the shelf, regarded as the navigation end pose in the task area. The robot's orientation is also randomized, but we ensure that the lamp remains visible to the RGB-D camera.

### D.1.2. REAL-TIME ADAPTABILITY TO DYNAMIC ENVIRONMENTS

As shown in Fig. 13, we show two trajectories of the chair. The first row shows the result of pushing the chair in a straight line, where, as can be seen in the right image, the prediction follows the chair as it moves. The second row shows the result of spinning the chair, and we can see that the prediction rotates together with the chair. This demonstrates the adaptability to non-static scene of the N2M module that it can adapt its predictions in real-time according to changing environments.

### D.1.3. GENERALIZATION TO UNSEEN SCENARIOS

For the four tasks shown in Fig 11(b-e), we qualitatively demonstrate N2M's remarkable data efficiency and generalizability along with its real-time performance. Note that we do not train manipulation policies for these tasks. When collecting rollouts for N2M training, we determine the preferable initial pose following our manual rule: the base is positioned approximately 0.5m away from the target object and oriented to face it. Although this setup does not employ a trained manipulation policy, these manual heuristic serves as the qualitative success criterion. We emphasize that these experiments are valid for qualitative analysis, aiming to provide a comprehensive evaluation of our proposed method's adaptability and robustness.

We collect 6, 12, 6, and 15 rollouts for the tasks of *Open Microwave*, *Use Laptop*, *Push Chair*, and *Toybox Handover*, respectively, with object pose and orientation randomized within a $3 \times 6$ m room. The N2M module is then trained with these rollouts and evaluated qualitatively across various environments, including ones unseen during training. We visualize the preferable initial poses predicted by our N2M module in Fig. 14. All the images were captured based on the predictions of the N2M module. To demonstrate the adaptiveness of our method, we overlaid multiple predictions into a single image for direct comparison.

In Fig. 14(a-d), we evaluate the N2M module in the same environment where the rollouts used for training the N2M module are collected. The N2M module successfully predicts poses that face the object from a distance of roughly 0.5m. Especially in Fig. 14(d), the N2M module's prediction adjusts the torso height according to the height of the toybox being held by the person.

We directly deploy the same N2M module trained for the *Toybox Handover* task in four entirely unseen environments, shown in Fig. 14(e-h). In particular, we qualitatively observe that the module consistently predicts appropriate adjustments in position, orientation, and torso height based on the toybox's location and orientation. Notably, this level of generalization is achieved with only 15 rollouts collected, demonstrating N2M's remarkable data efficiency and generalizability.

Finally, we demonstrate N2M's ability to adapt predictions in real-time based on environmental changes on the *Push Chair* task shown in Fig 11(d). Figure 13 shows the predicted preferable initial poses as the chair slides across the floor. Since N2M directly predicts the preferable initial pose distribution from an ego-centric RGB point cloud with a single forward pass and without needing any historical or global information, it can generate real-time predictions in dynamic environments. In contrast, methods like Mobi-$\pi$ (Yang et al., 2025) and MoTo (Wu et al., 2025a), which require global scene reconstruction during inference, are less suitable for non-static environments.

## D.2. Ablation Studies

### D.2.1. EXPERIMENT SETTING

All ablation study is conducted with Pick-and-Place Counter to Cabinet task in Robocasa. We use 20 rollouts to train N2M across all settings and follow Appendix. B for other detailed inference settings.

### D.2.2. FURTHER ABLATION

We conducted an ablation study to analyze the impact of viewpoint augmentation on data efficiency and performance. Viewpoint augmentation is crucial for both metrics, as demonstrated in Fig. 15(a). Further analysis in Fig. 15(b) shows that performance improves with a greater number of augmented views, saturating around $M = 300$ views. Fig. 15(c) establishes viewpoint augmentation as essential for high performance. Although we compare different augmentation methods, we note that viewpoint augmentation is complementary with positional augmentation and random noise. The highest performance attained when all augmentations are combined, demonstrating the synergy of these augmentations.

We conducted a further ablation study on our point cloud encoder choice and the use of pretrained weights. Although the PointBERT (Yu et al., 2022) encoder with pretrained weights achieved the highest performance (as shown in Fig. 16), the choice of encoder (PointBERT vs PointNet++ (Qi et al., 2017)) and use of pretrained weights did not critically affect the overall performance of N2M.

For the low performance of PointTransformerV3, as observed in Figure 16(b), we attributed this to an inherent architectural mismatch with our specific task. First, PointTransformerV3's reliance on extracting features from relative positions ensures translation invariance. This characteristic made it inappropriate for our method, which requires predicting an absolute position. We added positional encoding based on the absolute position of the points to address this issue. An additional challenge was PointTransformerV3's output feature size, which is dependent on the voxel grid size. This led to inconsistent output feature sizes across varying input point cloud shapes. To address this, we simply calculated the maximum value across the features to achieve size matching. Despite these algorithmic and structural compensations, we attribute the observed low performance of PointTransformerV3 to the inherent mismatch between its design purpose and the requirements of our task.

### D.3. Learned Representations

To visualize where the model focuses, we calculate the similarity between the output features of each token with the feature of a learned [cls] token used in Point-BERT. As shown in Fig 9, the model learns to focus on the salient regions, which aligns with the strong performance of the N2M module. We include additional visualizations in Fig 17.

### D.4. Failure Analysis

We provide an explanation regarding factors that may affect the performance of N2M.

**Small objects:** Since we downsample the point cloud before providing it to the model, we observe that objects as small as a pen or an eraser are typically indistinguishable, leading to erroneous predictions.

**Far distance:** False predictions occur when the robot is positioned too far from the region of interest. This distance causes the region to become indistinguishable and, crucially, pushes the input outside the viewpoint sampling area used for augmentation, making it an out-of-distribution scenario.

**Noisy sensor:** Noisy estimates of the point cloud generally made the point cloud out of distribution, leading to noisy predictions. To resolve this, we used a high-performance ZED2i camera to observe high-quality point clouds.

**Manipulation policy limits:** This represents the most critical failure case observed, and it is highly relevant to the core motivation of our project. Even when our module provided a reliable prediction, we still observed manipulation failure due to the extreme sensitivity and limited capability of the underlying policy.

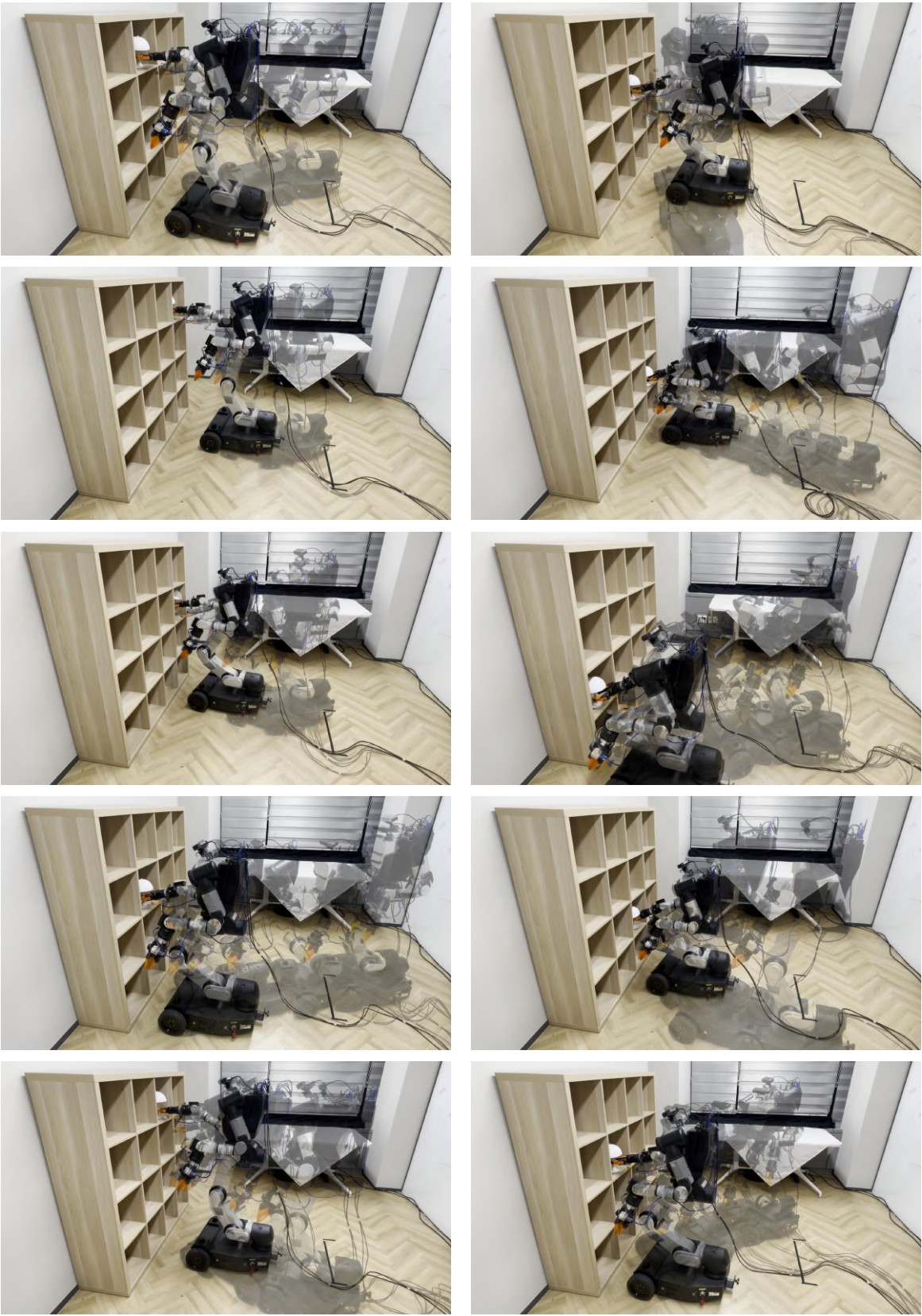

*Figure 12.* **Viewpoint robustness**. N2M can give a reliable base pose prediction at different viewpoints. We show ten consecutive successful executions of the *Lamp Retrieval* task to demonstrate this feature.

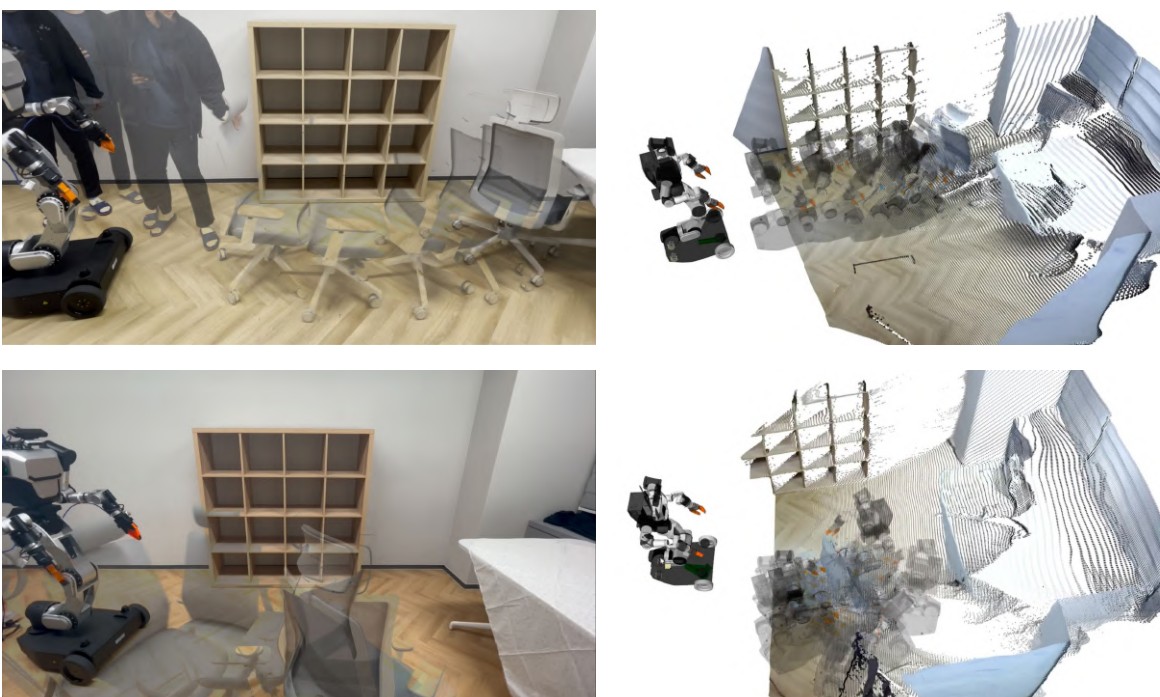

Figure 13. **Real-time adaptability**. N2M updates its predictions in real time in response to environmental changes. We show predictions in the *Push Chair* task, where the target chair slides (top) and spins (bottom) swiftly, and N2M can give real-time predictions.

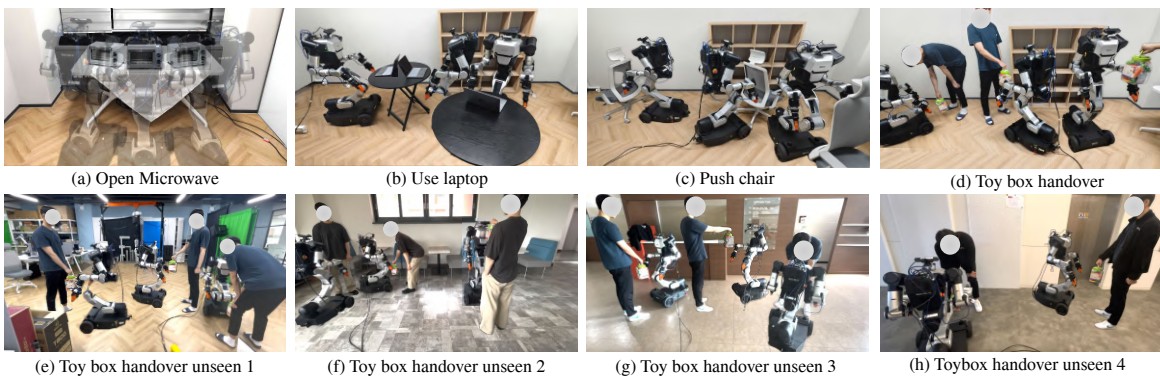

| (a) Open Microwave | (b) Use laptop | (c) Push chair | (d) Toy box handover |
| (e) Toy box handover unseen 1 | (f) Toy box handover unseen 2 | (g) Toy box handover unseen 3 | (h) Toybox handover unseen 4 |

Figure 14. **Generalization across scenarios**. (a-d) Predictions for unseen object placements within the training environment on four tasks described in Appendix C. (e-h) Zero-shot generalization to entirely unseen environments (different backgrounds and humans) in the *Toybox Handover* task.

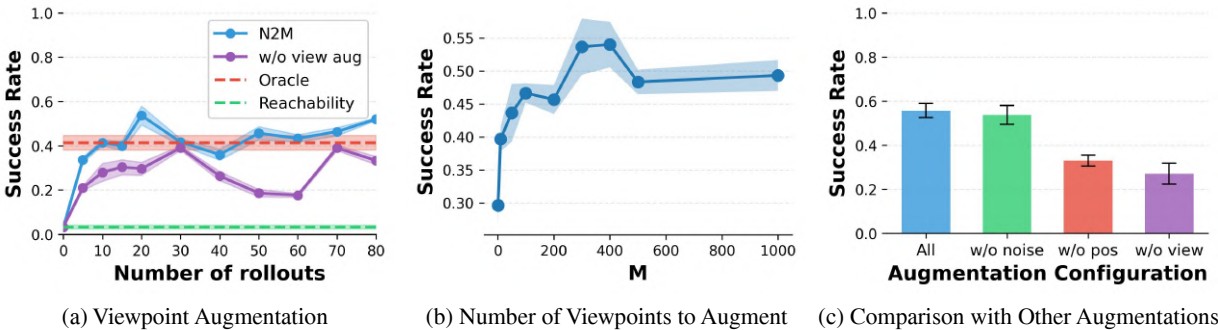

| (a) Viewpoint Augmentation | (b) Number of Viewpoints to Augment | (c) Comparison with Other Augmentations |

Figure 15. **Data augmentation ablation**. (a) Impact of different augmentation strategies on success rate. (b) Performance gain as the number of augmented viewpoints ($M$) increases, saturating around $M = 300$.

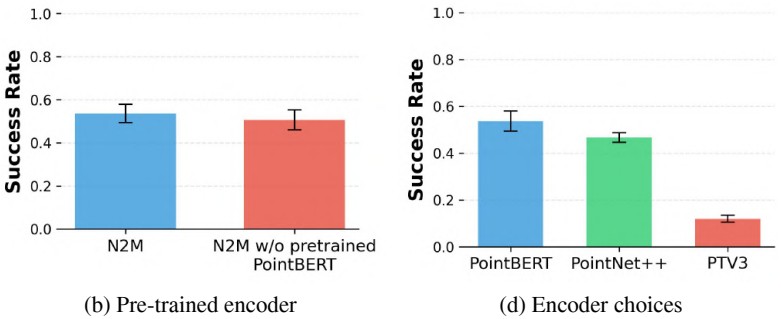

(b) Pre-trained encoder      (d) Encoder choices

*Figure 16.* **Encoder architecture ablation**. (a) Effect of utilizing pre-trained PointBERT weights. (b) Performance comparison across different point cloud backbones (PointBERT, PointNet++, PTV3).

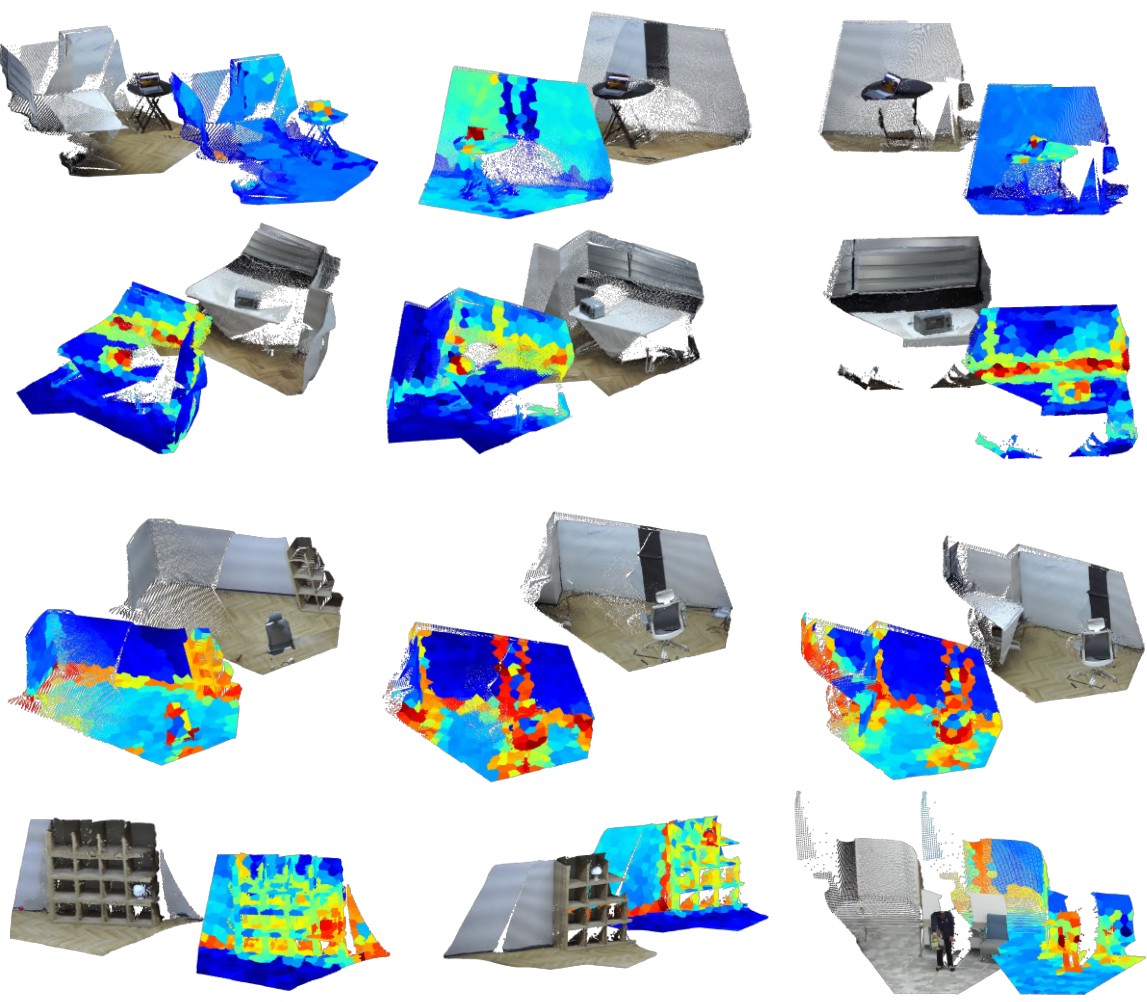

*Figure 17.* **Learned representation visualization**. Attention heatmaps demonstrate that the encoder focuses on task-relevant regions (e.g., the laptop, chair, microwave, lamp, and human) while ignoring background clutter. The tasks are described in Appendix C.

