# OpenReview forum: "N2M: Bridging Navigation and Manipulation by Learning Pose Preference from Rollout"
_ICML.cc/2026/Conference — ICML 2026 regular_

### Official Review · Reviewer_4aWZ · 2026-03-11

**Soundness:** 3
**Presentation:** 3
**Significance:** 2
**Originality:** 2
**Overall Recommendation:** 3
**Confidence:** 4

**Summary:**

This paper proposes N2M , a policy-aware framework designed to learn the inherent spatial preferences of the manipulation policy, guiding the robot base to the most favorable position to maximize the overall task success rate. The authors observe that while modern learning-based manipulation policies are capable, they are highly sensitive to their initial starting poses. N2M predict a multimodal distribution representing the spatial distribution of policy-friendly initial poses  from an onboard ego-centric RGB point cloud. Ground truth data is collected via trial-and-error policy rollouts, and a specific viewpoint data augmentation strategy is introduced to improve data efficiency and robustness.

**Compliance With Llm Reviewing Policy:**

Affirmed.

**Final Justification:**

Thanks to the authors for the detailed rebuttal. The response has resolved most of my queries. However, I apologize for not raising this earlier, but I believe that an approach which does not improve the inherent capabilities of the policy itself provides a limited contribution. Moreover, the method is not sufficiently novel or impressive. Therefore, I will maintain my current score.

**Key Questions For Authors:**

1.The experimental results show a low success rate for human-selected poses. A more detailed quantitative analysis to explain these failures should be provided.
2.Single Gaussian kernel seems sufficient in most experiments. So the multi-modal modeling capability of N2M is under-challenged by the current task suite.  Conversely, increasing the number of kernels for simple unimodal tasks may lead to prediction instability.
3.A premise of this paper is that human intuition is unreliable for predicting policy preferences. However, the zero-shot real-world evaluations revert to human judgment based on manual rules and visual reasonableness. The authors should use more reasonable evaluation methods to verify zero-shot performance.

**Limitations:**

1.The label generation process is tightly coupled with the rollouts of specific pre-trained policies. Any fine-tuning, update, or replacement of the underlying manipulation policy could render the previously trained N2M model obsolete due to policy shift.
2.The current evaluation primarily focuses on single-step tasks. In practical scenarios, mobile robots are often required to execute long-horizon composite tasks. If a predicted pose is optimal for the current task places, but hinders the next action, the system’s effectiveness will diminish in continuous missions.

**Strengths And Weaknesses:**

Strengths:
1.The paper addresses a significant and practical pain point in mobile manipulation: reaching a target's vicinity does not guarantee that the manipulation policy will succeed. The core idea of learning a base positioning preference directly from policy rollouts is an effective and easy approach.
2. Modeling base pose preference as a GMM with parameters predicted from RGB point clouds is straightforward and agnostic to the inner workings of the manipulation policy.
3.The viewpoint data augmentation strategy is a clever and effective mechanism for expanding a small set of successful physical rollouts into a diverse training set. This significantly enhances the system's data efficiency and its robustness to varying navigation end poses.

Weaknesses:
1.Insufficient analysis of human intuition failure modes: The paper asserts a significant gap between human intuition and policy preferences. While it is briefly noted that humans struggle to determine optimal base poses without prior knowledge of the policy. The paper should quantify the exact differences between the human-chosen poses and the policy-preferred poses.
2.Idealized real-world viewpoint augmentation: The real-world environments tested are relatively clean, neglecting the challenges in complex real-world. Crucially, projecting a single ego-centric point cloud to novel viewpoints cannot synthesize hidden geometry that was occluded in the original camera frame. The paper lacks a necessary discussion on how these occlusion shadows and synthetic rendering artifacts impact the robustness of the Point-BERT encoder's feature representations.
3.Lack of ablation on GMM kernels: Modeling the multi-modal distribution of preferable poses using a GMM is a theoretically sound approach. However, the paper lacks sufficient empirical validation regarding the choice of the number of kernels.
4.Contradictory evaluation of real-World generalization: The paper presents qualitative results for zero-shot generalization in unseen real-world environments, it lacks quantitative evaluation. For the tasks presented in Figure 11, the authors state that they do not train manipulation policies for these tasks and instead evaluate success based on a manual rule. Relying on human visual judgment to assess the reasonableness of predicted poses directly contradicts the paper's argument that human intuition is an unreliable proxy for actual policy preference.

---

> ### Author Rebuttal · Authors · 2026-03-31
>
> Thank you for your detailed and constructive feedback. We have carefully addressed your concerns and questions below, and we will incorporate these discussions into the revised manuscript.
>
> ### **1. Insufficient analysis of human intuition failure modes (Weakness 1 and Question 1)**
>
> Based on our empirical observations, we have clarified the cause of failure. The exact difference is that the **policy’s success region** is narrower than the **distribution of poses humans intuitively evaluate** as ideal. While humans select positions based on geometric feasibility and an intuitive sense of "smoothness," the resulting performance gap reveals that human-centric reasoning is an insufficient proxy for a policy’s specific operational preferences. To provide a quantitative understanding, we **added statistics** on typical failure categories.
>
> ### **2. Idealized real-world viewpoint augmentation (Weakness 2)**
>
> We respectfully clarify a **factual misunderstanding** regarding our data augmentation process. During our training data preparation, we do not render from a single ego-centric point cloud. As detailed in Section 3.3.1 (Raw Data Collection), "Multiple RGB point cloud frames are captured and stitched together to reconstruct the local task area $S$." The viewpoint augmentation (Section 3.3.2) is performed by rendering point clouds from this comprehensively reconstructed 3D scene $S$, effectively resolving the issue of occlusion shadows and hidden geometry that would arise from a single frame.
>
>
> ### **3. Kernel Choice of GMM (Weakness 3 and Question 2)**
>
> We appreciate the reviewer for highlighting this technical detail regarding our GMM-based modeling. We would like to provide the following clarifications:
>
> - **Selection of GMM Kernels:** We treat the number of kernels as a **user-defined parameter** intended to represent significant modes such as the distinct "left" and "right" approach angles for the Close Drawer task (Figure 2). We avoid using multiple kernels within a single mode to prevent optimization instability, as the reviewer correctly noted. Furthermore, because we only utilize successful base positions, distinguishing subtle sub-modes within a single cluster remains challenging.
> - **Limitations and Future Work:** While we find this approach sufficient for many scenarios, we acknowledge its limitations. We believe that more sophisticated distribution modeling, such as **Diffusion Models**, could resolve the need for manual mode configuration and further improve performance.
>
> Despite this specific modeling choice, we believe our identification on the critical importance of policy-aware base positioning, our viewpoint augmentation strategy, and the empirical insights gathered throughout our experiments remains a novel. We hope the reviewer will consider these broader contributions alongside our recognition of the current modeling constraints.
>
>
> ### **4. Concerns regarding evaluation of real-world Zero-shot Generalization (Weakness 4 and Question 3)**
>
> We would like to offer the following clarifications regarding the scope and placement of our analyses:
>
> - **Quantitative Analysis of Zero-Shot Generalization:** We believe the generalization capability of N2M is rigorously demonstrated through the quantitative experiments in Section 5.3 (Figure 6) and Section 6.1 (Figure 7). These tests specifically evaluate performance on entirely unseen scenes, as well as generalization across unseen shelf configurations.
> - **Role of Qualitative Assessment:** While we believe our qualitative analysis provides broader understanding of the method's behavior, we acknowledge the inherent ambiguity of such assessments. For this reason, we have placed these evaluations in the Appendix and framed it as "qualitative" evaluation. We kindly ask the reviewer to consider these as supplementary applications rather than the primary metric for our technical contributions.
>
> ### **5. N2M module being tailored to a specific policy (Limitation 1)**
>
> We appreciate the reviewer’s rigorous evaluation. For a detailed response to this point, please refer to our second response to Reviewer o6j2, where we address these considerations in depth.
>
>
> ### **6. Limited application to Long-horizon tasks (Limitation 2)**
>
> We respectfully clarify that long-horizon tasks do not pose a fundamental problem for our approach. **If a task requires a single stationary base for the entire task**, N2M should be trained to predict initial poses that satisfy the reachability and preference requirements of the entire sequence. Alternatively, **if a task necessitates relocating the base** (e.g., between workstations), it should be treated as distinct subtasks and train separate modules accordingly, a strategy successfully demonstrated in our Section 6.2 experiments.
>
> We hope this clarify our findings. However, we would welcome further guidance on any specific aspects of the sensitivity analysis that the reviewer feels remain unaddressed.

---

> > ### Author Rebuttal · Reviewer_4aWZ · 2026-04-03
> >
> > Thanks to the authors for the detailed rebuttal. The response has successfully resolved most of my queries, convincing me of the work's value and soundness. I also agree that this approach is useful for bridging navigation and manipulation policies. However, I apologize for not raising this earlier, but I believe that an approach which does not improve the inherent capabilities of the policy itself provides a limited contribution. Moreover, the method is not sufficiently novel or impressive. Therefore, I will maintain my current score

---

> > > ### Author Response · Authors · 2026-04-04
> > >
> > > We are glad that our initial reply resolved most of your concerns, and we appreciate your acknowledgment of N2M's effectiveness in bridging navigation and manipulation. We offer the following clarifications regarding your remaining points:
> > > - (a) The claim that our method does not improve inherent policy capabilities, limiting its contribution.
> > > - (b) The claim that the method is not sufficiently novel or impressive.
> > >
> > > ### **1. To concern (a): Cannot improve policies' inherent capabilities**
> > >
> > > While N2M does not directly alter a policy's internal weights, we argue that this orthogonality is precisely its strength, offering a highly cost-effective and complementary approach to system-level performance.
> > >
> > > - **A Cost-Efficient Path to Task Success:** Improving a policy's inherent capability typically requires massive data scaling or deploying resource-heavy foundation models (e.g., VLMs, World Models). In contrast, N2M offers an exceptionally cheap performance multiplier. **For example:** In our real-world experiment (Section 6.1), training a SOTA VLA policy required over **600 teleoperation trajectories (2 full workdays)**. Yet, when deployed on a mobile base, the success rate plummeted to **6.7%** due to base pose shifts. By collecting just **6 successful rollouts, taking less than 15 minutes** including resets, N2M boosted the success rate to **85%**. [**[Click to see how robust our method is]**](https://nav2manip.github.io/videos/Robustness_lamp_x8_05k_web.mov). N2M establishes a robust performance floor, functioning as a plug-and-play module that is fully compatible with any future advancements in policy architectures.
> > > - **Guiding Inherent Policy Improvement:** N2M can actually be leveraged to improve a policy's inherent capabilities. By explicitly modeling where a policy succeeds and fails across a task area, N2M serves as a spatial performance indicator. This distribution can directly inform active learning—highlighting out-of-distribution regions to guide targeted data collection or RL exploration to cover the full potential state space.
> > >
> > > ### **2. To concern (b): Method is not sufficiently novel or impressive**
> > >
> > > We acknowledge that our network architecture (Point Cloud Encoder + MLP) is straightforward. However, we argue that **algorithmic complexity does not equate to scientific novelty**.
> > > - Our core contribution is the **fundamental paradigm shift**: reframing the base positioning problem from a geometric reachability search into a learnable policy-preference prediction. The simplicity of our architecture actually highlights the strength of this formulation. Despite its lightweight design, N2M significantly outperforms recent, highly complex baselines [1-5] (Fig 5, Fig 8, Table 1).
> > > - **Beyond the base positioning problem we are solving**, our broader **conceptual contribution** is proving that a visuomotor policy's extreme spatial sensitivity is highly predictable and learnable from very sparse rollout data. In this sense, N2M functions as a critical policy-evaluation technique that effectively solves the "last mile" deployment bottleneck for data-driven mobile manipulation.
> > >
> > > [1] Wu, Zhenyu, et al. "Moto: A zero-shot plug-in interaction-aware navigation for general mobile manipulation." CoRL. 2025.
> > > [2] Yang, Jingyun, et al. "Mobi-$\pi$: Mobilizing Your Robot Learning Policy." CoRL. 2025.
> > > [3] Wu, Zhenyu, et al. "Momanipvla: Transferring vision-language-action models for general mobile manipulation." CVPR. 2025.
> > > [4] Jauhri, Snehal, Jan Peters, and Georgia Chalvatzaki. "Robot learning of mobile manipulation with reachability behavior priors." IEEE Robotics and Automation Letters 7.3 (2022): 8399-8406.
> > > [5] N. Vahrenkamp, T. Asfour and R. Dillmann, "Robot placement based on reachability inversion," 2013 IEEE International Conference on Robotics and Automation, Karlsruhe, Germany, 2013, pp. 1970-1975, doi: 10.1109/ICRA.2013.6630839.

---

### Official Review · Reviewer_NgGj · 2026-03-12

**Soundness:** 2
**Presentation:** 3
**Significance:** 2
**Originality:** 3
**Overall Recommendation:** 5
**Confidence:** 4

**Summary:**

The paper proposes the N2M module, which aims to predict the optimal base position for a mobile robot by directly learning the preferences of the manipulation policy. Instead of relying on pre-built 3D scene reconstructions, N2M directly utilizes egocentric point clouds acquired by an onboard RGB-D camera to predict a GMM distribution of optimal initial poses. To mitigate the high cost of physical exploration in the real world, the authors introduce a viewpoint augmentation strategy that significantly improves data efficiency. The method demonstrates exceptional inference speed and zero-shot generalization capabilities across various simulated and real-world tasks.

**Compliance With Llm Reviewing Policy:**

Affirmed.

**Final Justification:**

Most of my concerns have been resolved. This work provides an interesting solution for selecting the manipulation execution location in mobile manipulation tasks. It is highly commendable for a robotics paper to rely so heavily on real-world robot experiments.

**Key Questions For Authors:**

1.	When rendering new viewpoints during Viewpoint Augmentation, how does the algorithm handle occlusions (e.g., un-scanned obstacles between the new viewpoint and the target object) to avoid generating physically invalid or invisible training samples?

2.	If the underlying manipulation policy is a multi-task VLA model, can the N2M architecture easily predict task-specific pose preferences simply by concatenating language features?

3.	Will the task of spending time choosing the so-called appropriate execution location lose its application value due to the rapid progress of the domain's foundation model?

**Limitations:**

The inherent limitations of operational strategies: The fundamental role of N2M is to "maximize strengths and minimize weaknesses," guiding the robot to where the strategy performs best. However, if the operational strategy itself has a too narrow preference distribution or serious flaws in its capabilities, the task will still fail even if N2M provides the optimal prediction. This means that N2M cannot replace improving the robustness of the strategy itself.

**Strengths And Weaknesses:**

Strengths:

1.	Disruptive Inference Efficiency: By bypassing complex scene reconstruction, N2M achieves extremely fast end-to-end inference, removing engineering hurdles for real-time mobile manipulation in dynamic environments.

2.	Clever Data Augmentation and Generalization: The Viewpoint Augmentation strategy allows the model to converge using only a small number of physical rollouts and demonstrates impressive viewpoint invariance (zero-shot generalization) in unseen scenes.

3.	Solid Real-World Validation: The paper goes beyond simulation by conducting closed-loop testing on a physical robot across multi-stage tasks, proving its capability to handle the complexity of the real physical world.

Weaknesses:

1.	Tight Coupling with Specific Manipulation Policies: N2M learns the preferences of the current specific policy. This implies that if the underlying manipulation policy is updated, the N2M module must collect new rollout data and undergo fine-tuning.

2.	Perception Defects for Small Objects due to Point Cloud Representation: To ensure processing speed, the model downsamples the point cloud (comparing encoders like PointBERT and PointNet++), which severely degrades the system's ability to perceive tiny or distant objects.

---

> ### Author Rebuttal · Authors · 2026-03-31
>
> Thank you for your detailed and constructive feedback. We have carefully addressed your concerns and questions below, and we will incorporate these discussions into the revised manuscript.
> ### **1. N2M module being tailored to a specific policy (Weakness 1)**
> We appreciate the reviewer’s rigorous evaluation. For a detailed response to this point, please refer to our second response to Reviewer o6j2, where we address these considerations in depth.
> ### **2. Problem of Point Clound Downsampling (Weakness 2)**
> We thank the reviewer for the rigorous point on point cloud downsampling. We would first like to clarify that the downsampling to 8,192 points is a specific requirement of the pre-trained PointBERT backbone we leveraged for its robust semantic features, rather than a compromise for processing speed.
> - **Empirical Sufficiency:** Our experimental results demonstrate that this resolution is sufficient for capturing the fine-grained object details necessary for the tasks evaluated in this work.
> - **Hyperparameter Flexibility:** The point count is a tunable hyperparameter. For tasks requiring extreme precision, the resolution can be increased by trading off the use of specific pre-trained weights.
> - **Impact of Pre-training:** Furthermore, our ablation study in Appendix D.2 suggests that while the pre-trained backbone is beneficial, its impact on overall performance is relatively minor. This implies that increasing resolution at the expense of pre-trained weights would likely not result in a significant degradation of system performance.
> ### **3. Visual and Physical Invalidity during Viewpoint Augmentation (Question 1)**
> We appreciate this insightful question regarding how we handle infeasible camera positions. We employ several mechanisms to filter these cases, as detailed below:
> - **Visibility:** We use the object’s position to filter out base placements where the target object is not visible. In simulation, we utilize the ground-truth object position. For real-world experiments, we estimate the object’s location to be 0.5m in front of the labeled robot pose at a height of 1m. Empirically, we have found this heuristic provides a reliable estimate for the visibility check.
> - **Occlusion:** While we do not employ an explicit criterion for filtering occlusions, our 2m × 2m randomization region is designed to minimize severe occlusions. In practice, we observed that significant occlusions occurred very rarely within this workspace.
> - **Physical Infeasibility:** Since we acquire a point cloud of the local scene, we utilize it to generate an occupancy grid map. This allows us to effectively filter out viewpoints that are physically invalid or in collision with the environment.
>
> ### **4. Integration with multi-task VLA using language features (Question 2)**
>
> Yes thank the reviewer for pointing this out! We think incorporating language-conditioned general affordance learning for base positioning is a promising avenue to extend the capabilities of our framework. By transitioning toward a language-conditioned approach, the system could achieve greater generalization across diverse tasks and instructions.
>
> ### **5. Significance regarding rapid progress of foundation models (Question 3)**
>
> While we appreciate the reviewer’s perspective on foundation models, we believe our approach is an orthogonal contribution that remains significant regardless of a base policy's proficiency. Achieving uniform performance across the entire state space is realistically unattainable; even as policies advance, they inevitably exhibit "preference" regions where performance is optimized. For instance, while a foundation model may achieve a high success rate, metrics such as execution speed, motion smoothness, and throughput will still vary based on the initial configuration. Because this inherent non-uniformity persists even in advanced models, identifying high-performance regions is essential for extracting optimal behavior. Our work addresses this as a complementary optimization layer, ensuring long-term relevance by maximizing efficiency and reliability regardless of the underlying model's scale.
>
> ### **6. Inheritance of Manipulation Policy's Performance (Limitation)**
>
> As reviewer has pointed out, our N2M module is tightly coupled with the performance of manipulation policy. Although a policy can be optimized, it cannot fundamentally exceed its own performance boundaries. While N2M inherits manipulation policy's inherent failure modes, these components represent orthogonal research directions. Consequently, achieving higher overall task proficiency requires the simultaneous advancement of both the manipulation policy and the N2M module. Furthermore, N2M can serve as an indicator of regional policy proficiency, providing an avenue for active learning and targeted data collection to iteratively extend the policy's capabilities.

---

> > ### Author Rebuttal · Reviewer_NgGj · 2026-04-02
> >
> > I thank the authors for their detailed rebuttal. Most of my concerns have been resolved. This work provides an interesting solution for selecting the manipulation execution location in mobile manipulation tasks. Although the experimental setup remains relatively simple and lacks navigation validation in long-horizon manipulation tasks or larger spatial environments, I find it highly commendable for a robotics paper to rely so heavily on real-world robot experiments. Therefore, I am glad to **raise my score from 4 to 5**.
> >
> > I also recommend that the authors provide more objective evidence, such as new experimental results or citations from published papers, to address the other reviewers' questions, rather than relying on logical analysis.

---

> > > ### Author Response · Authors · 2026-04-02
> > >
> > > Thank you so much for recognizing our work, especially the experimental part! We included experiments across 11 tasks, 1 benchmark, 5 ablation studies, and 1 representation-level visualization analysis in our paper, just to try our best to deliver our simple but effective idea to the readers.
> > >
> > > We also want to thank you for your thoughtful questions and constructive suggestions regarding the rest of our rebuttal. Due to strict text limitations, we had to heavily condense our replies to ensure we could respond to every concern. We are more than happy to provide additional information and support once we receive any further questions from the reviewers.
> > >
> > > Thank you again for your great engagement and valuable time!

---

### Official Review · Reviewer_o6j2 · 2026-03-14

**Soundness:** 2
**Presentation:** 3
**Significance:** 2
**Originality:** 2
**Overall Recommendation:** 3
**Confidence:** 3

**Summary:**

The authors propose a new module for selecting the optimal base position from which to initiate a manipulation task. Specifically, the module takes an egocentric point cloud as input and outputs optimal locations, modeled as a Gaussian Mixture Model to account for the multi-modal nature of feasible positions. The prediction model itself is a linear-probed Point Transformer. Two data innovations are incorporated: (1) rollout-based sampling to collect ⟨observation, initial pose, success/failure⟩ triplets, and (2) NVS-based augmentation to enhance multi-view generalization capability.

**Compliance With Llm Reviewing Policy:**

Affirmed.

**Final Justification:**

The new visualizations help. However, my core concern remains unresolved: N2M requires substantial data collection and retraining for every new task-policy-environment-embodiment combination. The authors frame this as an intentional design choice, but it is a structural limitation. Calling it "intentional" does not make it less restrictive.

On data efficiency: the headline "6–10 samples" figure is compelling, but Section 5.1 acknowledges training multiple models and selecting the best — this selection cost is not accounted for in the efficiency claim.
On the reachability baseline: The suggested margin-augmented baseline was not tested. Plz have a refer to the CBF formation for deriving a boundary margin.

**Key Questions For Authors:**

- Is the current implementation of the reachability-based method a widely adopted practice? Would a simple spatial margin boost the success rate?
- Can we have a visualization of the success-rate distribution across candidate base poses for the reachability-based sampling?
- Are there other alternatives among non-policy-aware methods that should be considered?

**Strengths And Weaknesses:**

Strengths

- Policy-aware base positioning for manipulation is a previously under-explored domain. Typically, prior work bridges the two stages through rather formalized geometric methods.
- NVS-based data augmentation noticeably enriches the training distribution and is demonstrated to enhance model generalization capability.
- It is great to have extensive experiments spanning both simulation and real-world tasks.

Weaknesses

- The very first claimed contribution of the paper is "With experiments, we show that this misalignment stems from the manipulation policy's extreme sensitivity to its initial pose." I do not believe this is shown anywhere in the paper.
- The central narrative of the paper is that the positioning algorithm should be aware of the policy. Yet the proposed method itself takes neither the task specification nor the policy as input. Due to the ambiguity of task intention, policy heterogeneity, and embodiment heterogeneity, the trained model is constrained to a single task with a single policy tailored for a single embodiment. This is further confirmed by the acknowledgement in Section 5.1: "We train multiple N2M modules using 5 to 70 successful rollouts and select the best-performing model for each setting." In essence, the proposed paradigm simply offloads runtime cost to data collection and model training, yet in a less efficient manner.
- The reachability-based baseline is somewhat deliberately weakened and is simply too weak a comparator. A more fine-grained and stricter modeling of reachability could easily boost the baseline performance — for example, by incorporating a geometric margin. Compared with the heavy offline cost noted in the previous weakness, such policy-agnostic methods are both data-cheap and computationally efficient.

---

> ### Author Rebuttal · Authors · 2026-03-31
>
> ### **1. Lack support and visualization for policy's sensitivity (Weakness 1 and Question 2)**
> To address this, we have updated our manuscript and project website to better highlight both our existing data and new visualizations:
> - **Add more explanations to existing result**: In our manuscript, we have provided extensive quantitative baselines (e.g., Figures 5 and 7, and Table 1). The significant performance gap between different positioning strategies—namely reachability, N2M, and the Oracle—underscores that the manipulation policy is highly sensitive to different initial positions. To ensure this connection is immediately clear to the reader, we have newly added explanations to the paper that explicitly emphasize it shows sensitivity.
> - **Add Qualitative Visualizations**: To further support our claim, and directly following your suggestion, we newly added three figures to visualize how narrow the preferable initial pose distribution actually is. [**[Click to see]**](https://nav2manip.github.io)
>
> We hope these points clarify our findings. However, we would welcome further guidance on any specific aspects of the sensitivity analysis that the reviewer feels remain unaddressed.
>
> ### **2. N2M module being tailored to a specific policy (Weakness 2)**
> Your observation is absolutely correct: our current instantiation trains a module tailored to a specific policy. We agree this is a limitation for immediate multi-policy generalization, and we have updated our Limitations section to explicitly discuss this. However, **this focus was an intentional design choice**:
> - On one hand, as emphasized throughout the paper, a policy's spatial preference is fundamentally determined by the policy itself (e.g., its training data distribution and architecture). Therefore, to achieve optimal performance, the positioning module should be tightly coupled to the target policy. This design choice is precisely what enables us to achieve significantly higher success rates compared to all other non-policy-aware baselines.
> - On the other hand, this does not mean the N2M architecture is inherently restricted to a single policy. One potential method to achieve multi-policy generalization is to use the task name as a conditioning variable. However, rather than diluting our focus, we deliberately scoped this paper to rigorously validate N2M's core features and performance.
>
> ### **3. Stricter modeling of reachability can boost the performance (Weakness 3 and Question 1)**
> While reachability-based methods and heuristically modeling a margin may seem like reasonable choices for certain scenarios, we highlight two major limitations inherent to these approaches:
> - **Misalignment with policy's preference:** Even "reasonable" margins fail to capture a policy's specific operational needs. Our Human Intuition baseline (Section 6.2) demonstrates this; despite human annotators choosing geometrically feasible poses with sensible margins, a significant performance gap remains, proving that geometric proxies are unreliable.
> - **Restricted Scalability and Task Complexity:** These methods rely on precise object localization, which is often infeasible for articulated, deformable, or amorphous objects. In cluttered or complex scenes, defining stable geometric references for a margin becomes mathematically ill-posed.
>
> These limitations underscore the necessity for a method that directly reflects policy preference without being tethered to simple geometries or explicit perception. By predicting the initial base position directly from the point cloud, our pipeline bypasses the bottlenecks of explicit pose estimation, offering a robust and generalizable solution for diverse, realistic manipulation scenarios.
>
> ### **4. Reachability is not well implemented (Question 3)**
> We appreciate the reviewer’s rigorous evaluation of our reachability baseline. To clarify our implementation: we utilize **Pinocchio**, a standard kinematics library, to compute **inverse kinematics** (IK) based on the robot's URDF model and the target object's pose. There are two typical ways to utilize IK for reachability checks—pre-computing an Inverse Reachability Map, or using IK as a post-sampling filter. We adopted the latter approach. Specifically, we first sample collision-free base pose candidates and then run IK to verify if the object is kinematically reachable from each pose.
>
> To address the need for a stricter or more optimized criterion, we incorporated the **Human Intuition baseline** in Section 6.2. In this setup, human annotators select poses that are both geometrically feasible and strategically advantageous. Despite this being a significantly more sophisticated criterion than standard reachability, N2M still demonstrates superior performance. This remaining gap substantiates the extreme sensitivity of manipulation policies to their initial poses, highlighting the necessity of a policy-aware strategy that captures preferences beyond simple geometric or intuitive feasibility.

---

> > ### Author Rebuttal · Reviewer_o6j2 · 2026-04-03
> >
> > Thanks for the detailed rebuttal. The new visualizations and updated Limitations section are helpful improvements.
> >
> > That said, my core concern stands. N2M trades runtime cost for heavy offline sampling and per-task training, with no demonstrated generalization. Framing single-policy coupling as intentional reinforces rather than addresses the scalability problem.
> >
> > I maintain my original rating.

---

> > > ### Author Response · Authors · 2026-04-04
> > >
> > > We appreciate your acknowledgment that our response addressed your questions regarding policy sensitivity and baseline implementation. Based on your follow-up, it appears your remaining concerns:
> > > - (a) Our pamethod fails to show **generalizability**. Our method trades runtime cost for **heavy** sampling.
> > > - (b) And our method can only work with **single policy**. And our work avoids addressing **scalability problem**. It uses **offline samples** for training.
> > >
> > > Here is our further response:
> > >
> > > ### **1. To concern (a): Generalizability and Data Efficiency**
> > > We **respectfully disagree** that N2M lacks generalizability or relies on "heavy" sampling. In fact, generalizability and data efficiency are central features of our method, supported by both quantitative and qualitative evidence:
> > >
> > > **Generalizability**
> > > We demonstrated robust generalization across unseen object placements (Fig 7) and entirely unseen scenarios (Fig 6, Fig 14).
> > > * Please note that our experimental settings are **not toy** setting; they involve drastic environmental changes.
> > > * This generalization is not the result of **cherry-picking**. We provided videos showing **consecutive successes** to prove N2M works reliably even in completely unseen scenarios. [**Click to see**](https://nav2manip.github.io/).
> > >
> > > **Data Efficiency**
> > > This feature is highlighted throughout the paper:
> > > * Section 5.2 (Fig 5c) shows that with **only 10 samples**, N2M learns to predict an initial pose that improves the success rate from ~0% to ~40% (matching the Oracle).
> > > * In our real-world experiments (Fig 7c), we achieve near-optimal success rates with just **6 samples**.
> > > * In the appendix, we demonstrate that **15 samples** collected in one room (Fig 14d) generalize exceptionally well to totally unseen scenarios (Fig 14e-h). [**Click to see**](https://nav2manip.github.io/videos/Generalizability_all_in_one_x8_05k_web.mp4).
> > >
> > > Given this evidence, we respectfully request clarification on why our method is perceived as having limited generalizability or a heavy reliance on training data.
> > >
> > > ### **2. To concern (b): Single Policy, Offline Data, and Scalability**
> > >
> > > **Single Policy & Offline Data**
> > > We acknowledge that the current instantiation of N2M predicts initial poses for a single policy trained on offline data. However, these are **application-level** implementation choices, **not limitations of our core contribution**.
> > >
> > > Our **core contribution** is demonstrating that a data-driven policy's sensitivity to its initial pose is learnable from a very small amount of rollout data. Surrounding this insight, we designed a simple, effective method that significantly outperforms existing approaches without relying on additional assumptions, while offering fast inference and wide applicability. As a method-oriented conference paper, our objective is to rigorously validate this core claim rather than to present an exhaustive, multi-task framework.
> > >
> > > **Scalability**
> > > We **have** scalability-related experiments shown in Fig 5c and Fig 6 by incrementally increasing the amount of training data.
> > > * In both experiments, performance saturates very quickly due to our method's high data efficiency.
> > > * Fig 6 demonstrates that this rapid saturation is not overfitting; N2M successfully learns the general pattern of the task, allowing it to generalize across unseen textures and furniture layouts.
> > >
> > > We didn't scale this framework to multi-task or multi-policy domains, because it requires significant industrial-scale data collection, which **falls outside the scope of our core claim and contribution**.
> > >
> > > ### **3. Addressing the Final Assessment**
> > >
> > > Our paper received **"Fair"** scores for **Originality, Significance, and Soundness**, leading to a **Weak Reject**. We fully respect your decision, but based on the feedback provided, we are struggling to understand the justification for these scores.
> > >
> > > **Significance**: The community is rapidly moving from table-top to mobile manipulation. While traditional planning-based methods are robust to initial poses (given reachability), modern Behavior Cloning (BC) methods—which are the current SOTA—are extremely sensitive to the base pose. N2M addresses a critical, emerging bottleneck in deploying learning-based policy on mobile manipulation system.
> > >
> > > **Originality**: We systematically redesign the paradigm for this problem, moving away from flawed geometric proxies and other policy-aware alternatives, and achieve state-of-the-art performance where previous methods fail. Our benchmark (Table 1) backup this claim clearly.
> > >
> > > **Soundness**: Explained in Point 2 above.
> > >
> > > ### **4. Summary**
> > >
> > > We are sincerely grateful for your time and feedback. However, we feel the current assessment relies heavily on a misunderstanding of our method's data requirements and generalizability. We would greatly appreciate more specific details on the remaining perceived flaws so we can properly address them in future revisions.

---

### Decision · Program_Chairs · 2026-04-30

**Decision:**

Accept (regular)

**Comment:**

This paper introduces a learning algorithm to position a mobile manipulator appropriately so as to maximize the probability of success of execution of a learned manipulation policy at a given task.

Overall, the reviews highlight that the problem is interesting and valuable, the system well-designed, and the real-world experiments promising. The weaknesses revolved around whether the classical geometry-based baseline was sufficiently strong, the data-efficiency of the algorithm, and the sensitivity of the algorithm. The dependence on a specific policy, in my opinion, is acceptable - the objective of this paper, as the authors mention, is to optimize the performance of a specific system, not learn a general solution.

Having read the reviews, paper, and the author responses, I believe the paper is promising. A more thorough evaluation with a geometric baseline, and more rigorous evaluations of the data efficiency would have made it stronger.